# Engineered hypermutation adapts cyanobacterial photosynthesis to combined high light and high temperature stress

Huili Sun[1,2,3,4,8], Guodong Luan [1,2,3,4,5,8] ✉, Yifan Ma[1,2,3,6], Wenjing Lou[1,2,3], Rongze Chen[1,2,3,4], Dandan Feng[1,2,3], Shanshan Zhang[1,2,3,4], Jiahui Sun[1,2,3,4] & Xuefeng Lu [1,2,3,4,5,7] ✉

Photosynthesis can be impaired by combined high light and high temperature (HLHT) stress. Obtaining HLHT tolerant photoautotrophs is laborious and time-consuming, and in most cases the underlying molecular mechanisms remain unclear. Here, we increase the mutation rates of cyanobacterium *Synechococcus elongatus* PCC 7942 by three orders of magnitude through combinatory perturbations of the genetic fidelity machinery and cultivation environment. Utilizing the hypermutation system, we isolate *Synechococcus* mutants with improved HLHT tolerance and identify genome mutations contributing to the adaptation process. A specific mutation located in the upstream non-coding region of the gene encoding a shikimate kinase results in enhanced expression of this gene. Overexpression of the shikimate kinase encoding gene in both *Synechococcus* and *Synechocystis* leads to improved HLHT tolerance. Transcriptome analysis indicates that the mutation remodels the photosynthetic chain and metabolism network in *Synechococcus*. Thus, mutations identified by the hypermutation system are useful for engineering cyanobacteria with improved HLHT tolerance.

Photosynthesis is the most important biochemical process on Earth; through this process, photoautotrophs convert solar energy and carbon dioxide into chemical energy and organic compounds. Photoautotrophs, primarily comprising plants and algae, provide the essential primary productivity for biosphere development and maintenance. Improving the efficiency and stability of photosynthesis has long been an important direction for plant breeding and crop improvement[1,2]. Photosynthesis is performed by complex and sophisticated biochemical and molecular machineries, and high photosynthetic efficiency could only be achieved within an optimal range of physical and chemical parameters, while extreme and fluctuating environmental conditions would impair photosynthetic efficiency[3]. During the agriculture-mode cultivation process, plants and algae are required to withstand multiple impairing climate factors, which, individually or in combination, disturb the photosynthesis activities and biomass production[4,5]. High light and high temperature are both severe environmental stressors that impair photosynthetic efficiency, and would sometimes act in tandem, causing superimposed effects[6–8]. Excessive exposure to solar energy and the resulting high temperature beyond the acclimation range of photosynthesis machineries would cause the accumulation of intracellular reactive oxygen species (ROS), which further induces photosystem impairments[9], impairs the protein

[1]Qingdao Institute of Bioenergy and Bioprocess Technology, Chinese Academy of Sciences, No. 189 Songling Road, 266101 Qingdao, Shandong, China. [2]Shandong Energy Institute, No. 189 Songling Road, 266101 Qingdao, Shandong, China. [3]Qingdao New Energy Shandong Laboratory, 266101 Qingdao, Shandong, China. [4]College of Life Science, University of Chinese Academy of Sciences, 100049 Beijing, China. [5]Dalian National Laboratory for Clean Energy, 116023 Dalian, Liaoning, China. [6]College of Life Science and Technology, Central South University of Forestry and Technology, 410004 Changsha, Hunan, China. [7]Laboratory for Marine Biology and Biotechnology, Qingdao National Laboratory for Marine Science and Technology, 266237 Qingdao, Shandong, China. [8]These authors contributed equally: Huili Sun, Guodong Luan. ✉e-mail: luangd@qibebt.ac.cn; lvxf@qibebt.ac.cn

synthesis and repair activities, and disrupts intracellular homeostasis[10]. Combined high light and high temperature (HLHT) stress poses threats to agriculture and other photosynthesis-derived economic activities and can happen more widely and frequently as the climate changes[1,6,7,11,12]; thus, improving the tolerances of photoautotrophs to HLHT is of great significance. However, although the specific responsive or protective mechanisms of plants and algae toward single stress (high light[13,14] or high temperature[15,16]) have been explored, understanding of tolerance mechanisms to the combination of the two stresses (HLHT) is still relatively limited.

Cyanobacteria are important model organisms for exploring and engineering photosynthetic mechanisms. Sharing a common ancestor with chloroplasts, cyanobacteria have similar photosynthetic mechanisms with eukaryotic microalgae and higher plants. Besides, cyanobacteria also have the characteristics of short life cycles, rapid growth rates, and simple structures, allowing convenient genotype-phenotype mapping for photosynthetic research[17]. Following a typical forward genetic research paradigm, genetic and phenotypic diversities could be more conveniently introduced or created into rapidly expanding cyanobacteria offspring populations, and the determining mechanisms and mutations for target photosynthesis-related phenotypes could be identified using multiple omics approaches[18]. However, the acquisition of mutants with desired phenotypes remains a bottleneck in this research route, especially for the optimization of complex physiological properties, such as HLHT tolerance. The conventional adaptive laboratory evolution (ALE) approach, even enhanced with physical and chemical mutagenesis treatments, usually requires weeks or months to obtain cyanobacteria mutants with improved tolerances to high light or high temperature stress[19–22]. Moreover, mutants with improved tolerance to the combined high-light and high-temperature stresses were rarely reported[23]; thus, the mechanisms of cyanobacterial photosynthesis tolerance to HLHT stress are still not as clearly understood as those for every single stress.

The generation of abundant genetic diversity in cyanobacteria offspring populations is a prerequisite for phenotypic evolution and isolation; however, their highly accurate genome replication machinery poses formidable obstacles[24]. Based on classical models and data, it can be estimated that it would take thousands of cell replication events to form one genetic mutation into the cyanobacteria genome[25]. Although the desired mutant could still be obtained from large-sized populations (from the culture with increased cell density or volume), the process for selecting and identifying the mutations would be time and labor consuming.

In this study, we use an artificial hypermutation system to accelerate genome mutagenesis for adapting photosynthesis to HLHT stress in cyanobacteria. We show that we can isolate *Synechococcus* mutants with improved HLHT tolerance within two weeks and identify functional mutations and mechanisms that determine the evolved traits using forward genetic analysis and reverse genetic confirmation. Our study reveals that enhanced expression of shikimate kinase, an essential enzyme in the shikimate pathway, plays an effective role in optimizing the efficiency and stability of photosynthesis in cyanobacteria.

## Results

### Disrupting the natural DNA replication fidelity mechanism to increase mutation rates

The high fidelity of genome replication processes in biological cells is ensured by multiple levels of proofreading and repair mechanisms[25], including base selection, 3'->5' exonuclease activity, methyl-directed mismatch repair (MMR), DNA damage repair, translesion DNA synthesis (TLS), etc. (Fig. 1). Inactivation of fidelity elements or overexpression of mutagenic (error-prone) elements increased mutation rates of genome replication in some model microorganisms[26,27]. However, the DNA replication mechanism and the associated genes are

not well understood in cyanobacteria. Therefore, a set of potential target genes (for deletion or overexpression) was selected by referring to bioinformatics annotation and replication mechanism for genomes of microorganisms[28] (Fig. 1 and Supplementary Table 1). Eight genes participating in the DNA replication fidelity process (*mutS*, *mutL*, *dam*, *xthA*, *uvrB*, *uvrC*, *mutY*, and *mutM*) were knocked out from the *Synechococcus* genome, and four genes contributing to error-prone DNA replication (*umuC*, *umuD*, *recN*, and *recA*) were overexpressed individually or in combination. As shown in Supplementary Fig. 1, cell growths of the recombinant strains were unaffected.

The generation frequencies of rifampicin-resistant colonies were used to calculate the genome replication mutation rates of *Synechococcus* (Supplementary Figs. 2 and 3)[27]. The relative mutation rates of five recombinant strains, HS1 (Δ*uvrB*), HS2 (Δ*uvrC*), HS10 (Δ*mutS*), HS7 (OE-*recA*; OE, overexpressing), and HS25 (OE-*umuDC*), were increased, and *mutS* inactivation led to the highest mutation rates (approximately 10-fold increase) in the HS10 strain (Fig. 1b). To further enhance the in vivo mutagenesis strengths, recombinant strains carrying multiple genetic modifications were constructed. As the HS10 strain showed the highest mutation rates, Δ*mutS* mutation was used in combination with the other four mutations, obtaining HS83 (Δ*mutS*-Δ*uvrB*), HS84 (Δ*mutS*::P$_{cpcB560}$-*recA*), HS88 (Δ*mutS*::P$_{cpcB560}$-*umuDC*), and HS96 (Δ*mutS*-Δ*uvrC*). Previously, it was reported that the combination of Δ*mutS* and Δ*dam* showed synergetic effects on elevating mutation rates in *E. coli*[29]. Thus, a similar recombinant strain, HS91 (Δ*mutS*-Δ*dam*) was also constructed. Further, the combination of TLS and NER was adopted by simultaneously overexpressing *umuDC* and inactivating *uvrB* or *uvrC* to generate strains HS89 (Δ*uvrB*::P$_{cpcB560}$-*umuDC*) and HS90 (Δ*uvrC*::P$_{cpcB560}$-*umuDC*), respectively. Among the second-round recombinant strains, HS84 (Δ*mutS*::P$_{cpcB560}$-*recA*) showed an over 130-fold increase in mutation rate (Fig. 1c), indicating that the two genes have interactive effects on *Synechococcus* DNA replication fidelity. Meanwhile, double knockout of *mutS* and *dam* showed synergetic effects on elevating mutation rates in *Synechococcus*, resulting in 21-fold increased mutation rates. In contrast, the other combinations with Δ*mutS* did not yield a significant increase in mutation rates. Overexpressing *umuDC* in *Synechococcus* strains with inactivated *uvrB* (HS89) or *uvrC* (HS90) also imparted additional mutagenic effects, resulting in 12- and 36-fold increased mutation rates, respectively. In contrast, the mutation rates could not be elevated further by additional manipulations with HS84 (Fig. 1c). Although the DNA replication mechanism and functions of the related genes are not clearly understood in *Synechococcus*, combinatorial knockout-overexpression manipulations of multiple putative genes enabled us to elevate the genome replication mutation rates by two orders of magnitude.

### Triggering the hypermutation state by imposing environmental stresses

Preliminary examinations demonstrated that the genome replication mutation rates in *Synechococcus* strains were also affected by the cultivation conditions (Supplementary Fig. 4a–c). It was observed that the mutation rates of most mutant strains with defective replication fidelity cultured at 33 °C and 250 μmol photons/m²/s (white fluorescent light) were elevated compared to those cultured at 30 °C and 150 μmol photons/m²/s (white fluorescent light). Therefore, to investigate the effects of light intensity and temperature on replication fidelity, the WT *Synechococcus* and HS84 strains (Δ*mutS*::P$_{cpcB560}$-*recA*) were cultivated in an MC1000 cultivator under controlled environmental conditions. As shown in Fig. 2, the mutation rates of both the WT and HS84 strains were regulated by temperature and light intensity. It was observed that the mutation rates in the WT cells cultivated at 30 °C were elevated by 57% by increasing the light intensities from 400 to 1000 μmol photons/m²/s, whereas a simultaneous increase in temperature to 42 °C would further increase the mutation rate by 3.8-fold of the initial level (Fig. 2a). The same trends were also

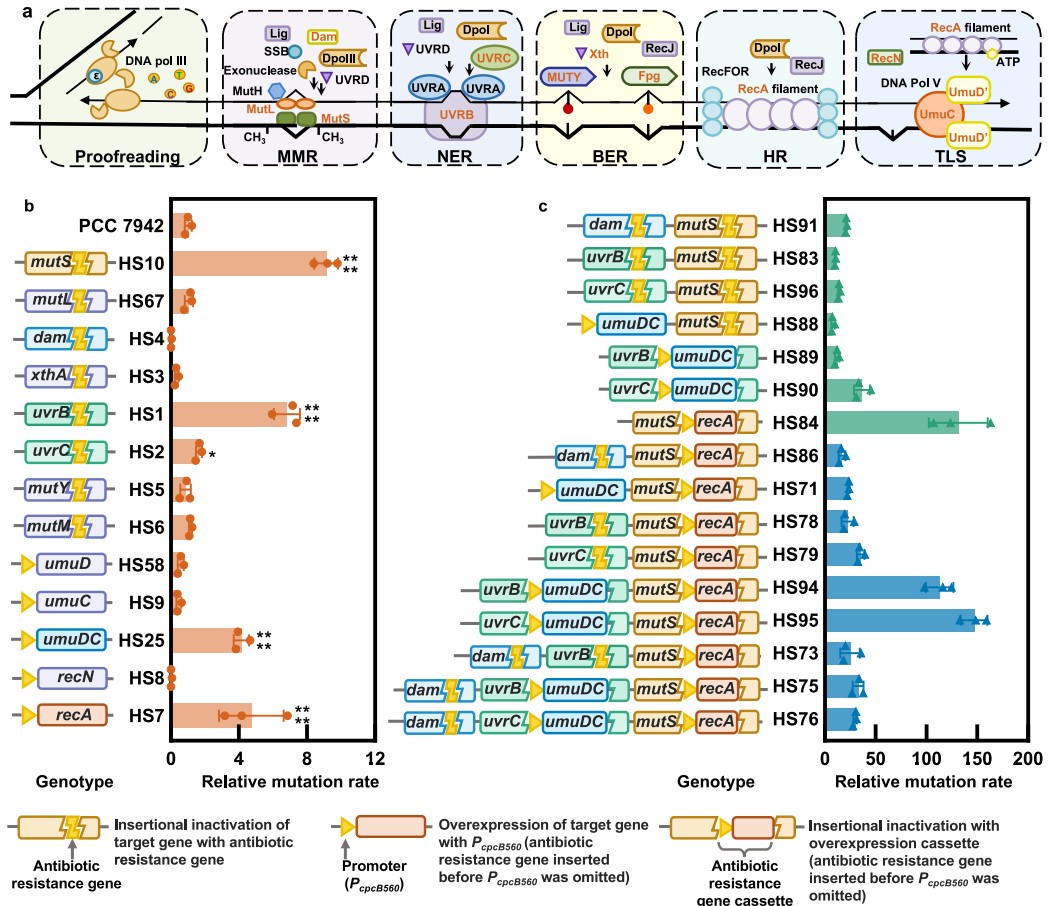

**Fig. 1 | Identification and selection of putative genes involved in DNA replication fidelity in *Synechococcus*. a** The schematic diagram of DNA replication fidelity-related mechanisms and the selected components for manipulation. From left to right: proofreading, methyl-directed mismatch repair (MMR; *mutS*, *mutL*, and *dam*), nucleotide excision repair (NER; *uvrB* and *uvrC*), base excision repair (BER; *mutM*, *mutY*, and *xthA*), homologous repair (HR; *recA*), translesion DNA synthesis (TLS; *umuD*, *umuC*, *recN*, and *recA*). Illustrated referring to the KEGG pathway (syf0303, syf03430, syf03420, syf03410, and syf03440) and a publication[71]. **b** The genotypes of *Synechococcus* mutants carrying a single inactivated/overexpressed DNA replication fidelity-related gene and the associated mutation rates relative to that of the wild-type (WT) control (To ensure consistency of comparison, the WT mean mutation rates of 12 biological replicates from 4 independent experiments were used as the denominator in the calculation of relative mutation rates.). Statistical analysis was performed using a two-tailed unpaired Student's *t*-test (*$p < 0.05$, ****$p < 0.0001$). From up to down: $p < 0.0001$, $p < 0.0001$, $p = 0.0242$, $p < 0.0001$, $p < 0.0001$. **c** The genotypes of *Synechococcus* mutants carrying multiple inactivated/overexpressed DNA replication fidelity-related genes and the associated mutation rates relative to that of the WT. The *Synechococcus* strains were cultured at 30 °C and 150 μmol photons/m²/s (white fluorescent light) until the logarithmic growth phase, and $10^8$–$10^9$ cells were coated on BG11 plates containing 15 μg/mL rifampicin. The frequencies of rifampicin-resistant colonies were utilized to evaluate the mutation rates of genome replication, and the relative mutation rates were calculated by comparing them with that of the WT. The experiments were replicated twice. Data are presented as mean values ± SD ($n = 3$ biological replicates). Source data are provided as a Source data file.

observed in HS84 strain. Cultivating the HS84 cells at 42 °C and 1000 μmol photons/m²/s increased the genome mutation rates up to 1200-fold higher than that of the WT (Fig. 2b). The results presented in Fig. 2b are the representative data from multiple replicates, while in some batches of replicates, the calculated replication rates reached over 8500-fold higher than that of the WT (Supplementary Fig. 4d).

Considering cyanobacterial photosynthetic metabolism, it could be speculated that the effects of environmental temperature and light intensity on the genome replication mutation rate may be due to the increased level of intracellular ROS generated during photosynthesis, which might cause genomic mutations through DNA damage. As shown in Fig. 2c, the ROS concentration was increased in HS84 cells under HLHT conditions. Thus, a genome mutation rate regulatory mechanism was proposed for *Synechococcus* (Fig. 2d), in which the mutations induced by the intracellular ROS (accumulated during photosynthesis) could be repaired by the replication fidelity machinery. The environmental stresses could affect the potential mutation-fidelity balance by disturbing photosynthesis or other physiological

activities and elevating the intracellular ROS levels. Thus, when essential fidelity mechanisms are inactivated, imposing non-lethal stress could drive the defective genomic replication process into a hypermutation state.

We compared the genomic mutagenesis effects of the engineered hypermutable system with classical chemical mutagenic agent methyl methanesulfonate (MMS), which has been used for inducing point-mutations in *Synechocystis* and promoting high-temperature adaptations[21]. As shown in Supplementary Fig. 5a, a 90% lethal rate (which was usually used in microbial random mutagenesis) was observed in samples treated with 2% (volume to volume) MMS. After recovery cultivation in fresh BG11, mutation rates of the treated cells were about 30 folds higher than those of the control (Supplementary Fig. 5b). Considering the advantages of genomic mutagenic strengths and biosafety (chemical mutagens are generally considered to be highly carcinogenic), the engineered hypermutable system could be a useful supplementation to the existing cyanobacterial mutagenesis approaches.

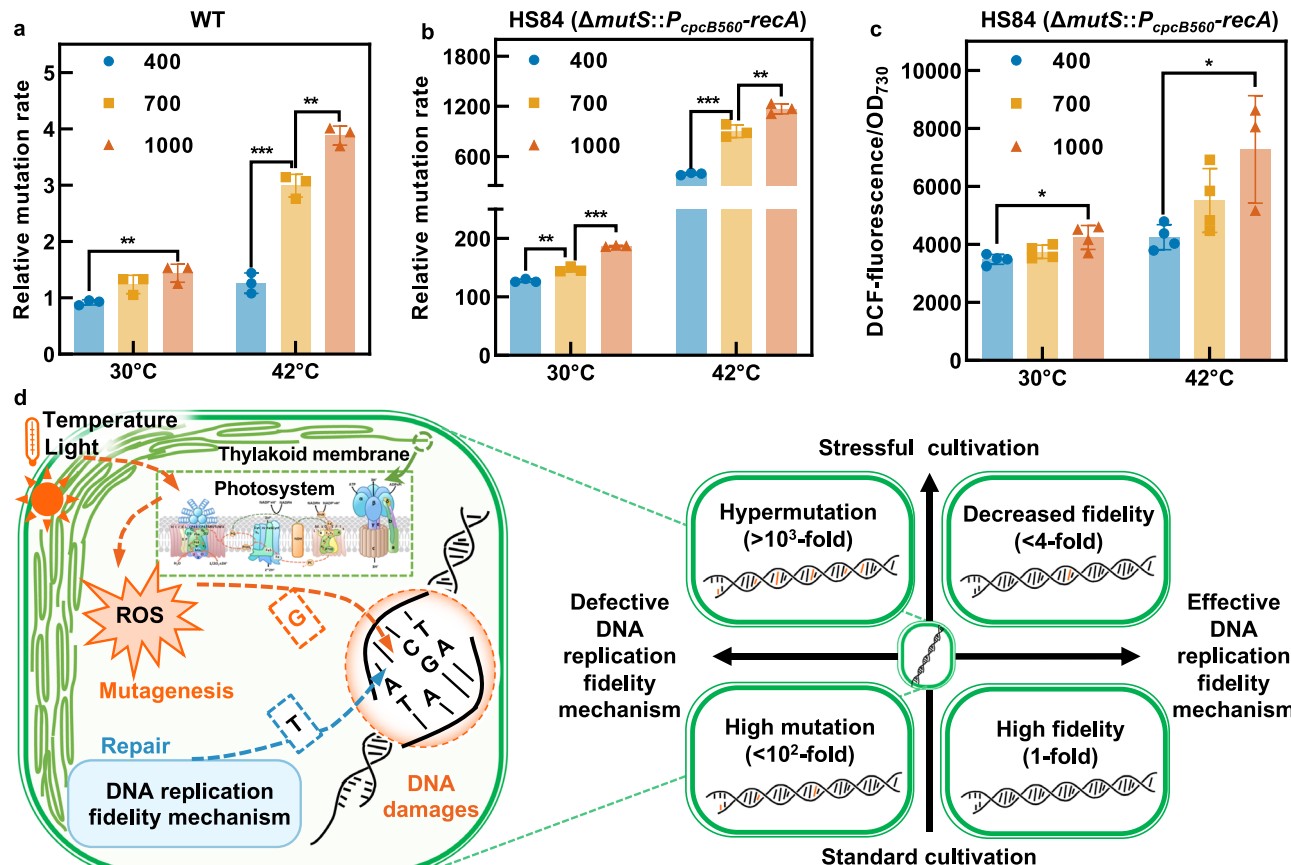

**Fig. 2 | Synergistic effects of inefficient replication fidelity mechanism and environmental stresses on genome replication mutation rates in *Synechococcus* cells. a** The relative mutation rates of WT *Synechococcus* cultivated under 30 °C and 42 °C with the light intensities of 400, 700, and 1000 μmol photons/m²/s. **b, c** The relative mutation rates (**b**) and ROS content (**c**) of HS84 strain cultivated under 30 °C and 42 °C with the light intensities of 400, 700, and 1000 μmol photons/m²/s. **d** The hypothesized mechanism model of regulating the mutation rates in *Synechococcus*. The horizontal axis represents the defective (left of the origin) and effective (right of the origin) state of the DNA replication fidelity machinery, and the vertical axis represents the normal (below the origin) and stressful (above the origin) cultivation conditions. When WT cells with functional replication fidelity mechanisms were cultivated under normal conditions, genome replication was highly accurate, and the mutation rates were considered to be at the basal level. Cultivation under unfavorable conditions (42 °C and 1000 μmol photons/m²/s) led

to an accumulation of ROS in WT cells and an increase in mutation rate (fourfold up-regulation). When mutant cells with defective replication fidelity mechanisms, were cultivated under normal conditions, the mutation rate was increased by 100-fold, owing to the error-prone DNA replication process. Cultivation under unfavorable conditions (HLHT stress) triggered a hypermutation state in the mutant cells and increased the relative mutation rate by over 1000-fold. Each experiment was replicated more than twice to ensure its reliability. Statistical analysis was performed using a two-tailed unpaired Student's *t* test (*$p < 0.05$, **$p < 0.01$, ***$p < 0.001$). The *p* values in **a**–**c** are (from left to right) 0.0059, 0.0004, 0.0042, 0.0031, 0.0001, 0.0003, 0.0083, 0.0145, and 0.0226. Data are presented as mean values ± SD. *n* = 3 biological replicates in **a**, **b**. All sample size *n* (biological replicates) in **c** is 4 except that *n* of the 42 °C and 1000 μmol photons/m²/s column is 3. Source data are provided as a Source data file.

## Improving HLHT tolerance using the engineered hypermutation system

*Synechococcus* offsprings generated by the hypermutation system were screened for improved HLHT tolerance. Following a classical forward genetics research paradigm (as shown in Fig. 3a), HS84 cells (Δ*mutS*::P$_{cpcB560}$-*recA*) from the hypermutation cultivation process were inoculated on BG11 plates and incubated at lethal temperature and illumination conditions (for the WT *Synechococcus* strain) for four days. WT *Synechococcus* cells subjected to the same cultivation conditions or treated with 2% MMS were also cultivated under the same selective process as controls. Multiple HS84 colonies were obtained under the selective cultivation conditions; however, WT colonies (with or without MMS mutagenesis treatment) were not obtained (Fig. 3a and Supplementary Fig. 6). The colonies generated in the selective environment were re-cultivated under the same conditions to confirm the tolerant phenotype. Finally, the selected candidate strains were genetically engineered to restore the genome replication fidelity machinery.

Three of the mutant strains/colonies were randomly selected and engineered to restore the *mutS* gene and delete the additional *recA* in a

complementation step, obtaining HS121, HS122, and HS123 strains (Supplementary Fig. 7). Growth assays demonstrated that the HLHT tolerances of the three mutant strains were significantly improved. Under normal conditions (30 °C and 500 μmol photons/m²/s), slight growth advantages in the three mutant strains were detected compared with the WT (Fig. 3b). Moreover, elevating the temperature to 42 °C at the same light intensity optimized growth of all the strains, while the evolved strains showed better growth than WT (Fig. 3c). Under stressful conditions with high temperature (45 °C and 500 μmol photons/m²/s) and high light intensity (42 °C and 1500 μmol photons/m²/s), the evolved strains showed obvious growth advantages, whereas the WT strain could hardly survive (Fig. 3d, e). Under more severe conditions (Fig. 3f, 42 °C and 2500 μmol photons/m²/s; Fig. 3g, 45 °C and 1500 μmol photons/m²/s), the evolved strains could still survive and grow, displaying optimized adaptability to HLHT stress. In addition, the improved tolerance phenotypes were maintained in the three evolved strains during long-term storage and cultivation. Furthermore, the relative mutation rates of the evolved strain were restored to the normal level, indicating that genotype complementation repaired the fidelity

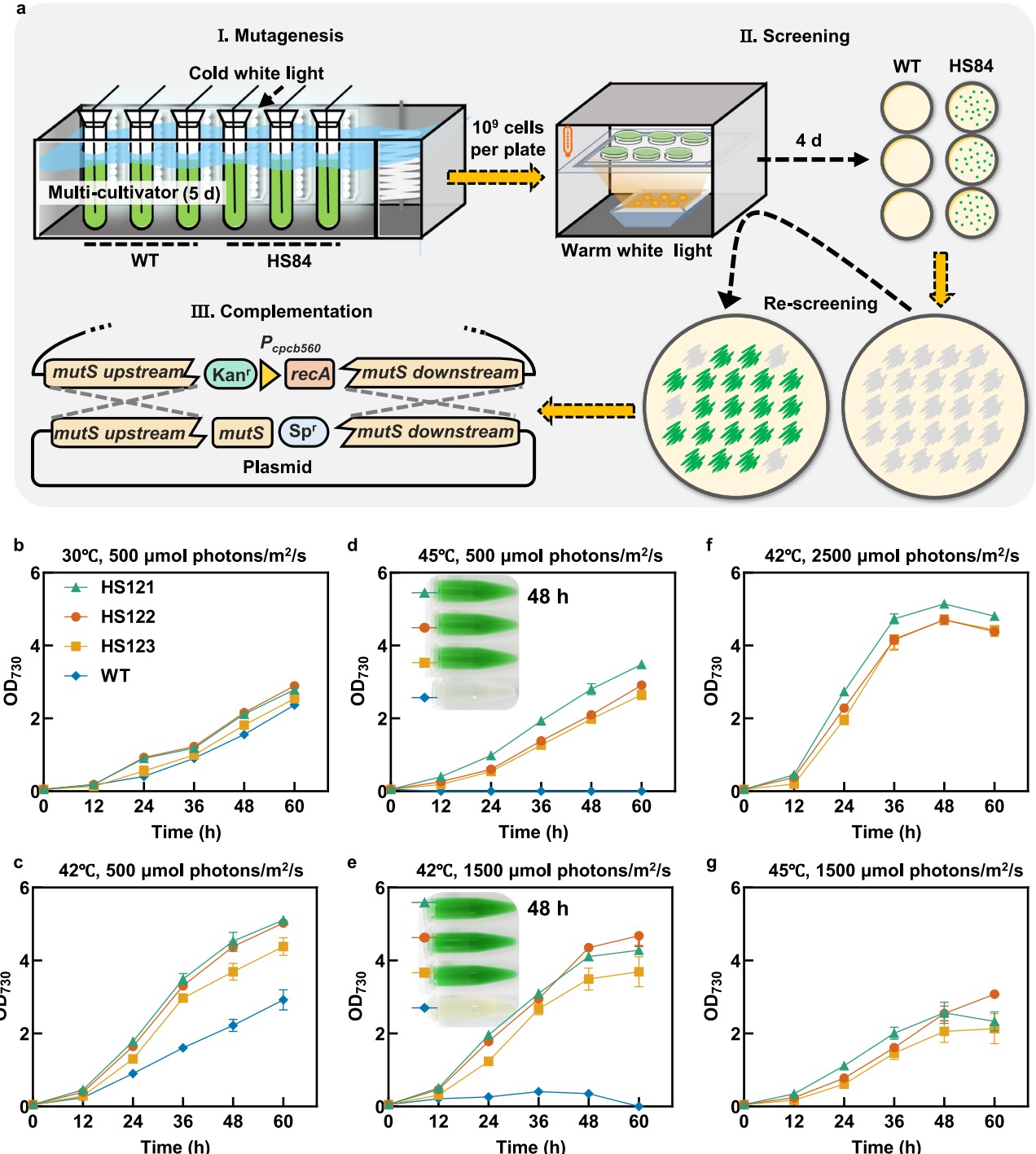

**Fig. 3 | Isolation of *Synechococcus* mutants with improved tolerances to HLHT stress using the hypermutation system. a** Scheme of the evolution and isolation process of the hypermutation system aiming to generate HLHT tolerant *Synechococcus* mutants. In the mutagenesis step (I), *Synechococcus* cells were cultured in a multi-cultivator at 42 °C and 1000 μmol photons/m²/s (cold white light). Approximately 10⁹ cells in the logarithmic growth phase were harvested and coated on BG11 plates and cultivated in an incubator at 44 °C and 500 μmol photons/m²/s (warm white light) for screening (step II, screening). After 4 days of incubation, colonies were inoculated on fresh BG11 plates for re-screening in the incubator. In the final step (III; complementation), the tolerant mutants (generated in the re-screening process), were genetically modified to restore the replication fidelity deficiency (by restoring the coding sequence of *mutS* and deleting the additional *recA*). The relative replication mutation rates of the three evolved strains (HS121, HS122, and HS123) isolated following the above procedures were calculated by determining the frequencies of the rifampicin-resistant colonies using the WT *Synechococcus* as control. **b**–**g** Growth rates of H121, HS122, and HS123 strains were evaluated using the multi-cultivator at 30 °C and 500 μmol photons/m²/s (**b**), 42 °C and 500 μmol photons/m²/s (**c**), 45 °C and 500 μmol photons/m²/s (**d**), 42 °C and 1500 μmol photons/m²/s (**d**), 42 °C and 2500 μmol photons/m²/s (**f**), and 45 °C and 1500 μmol photons/m²/s (**g**). The broths of the **d**, **e** samples were collected after 48 h of cultivation and were photographed. Each experiment was replicated more than twice to ensure its reliability. Data are presented as mean values ± SD (*n* = 3 biological replicates). Source data are provided as a Source data file.

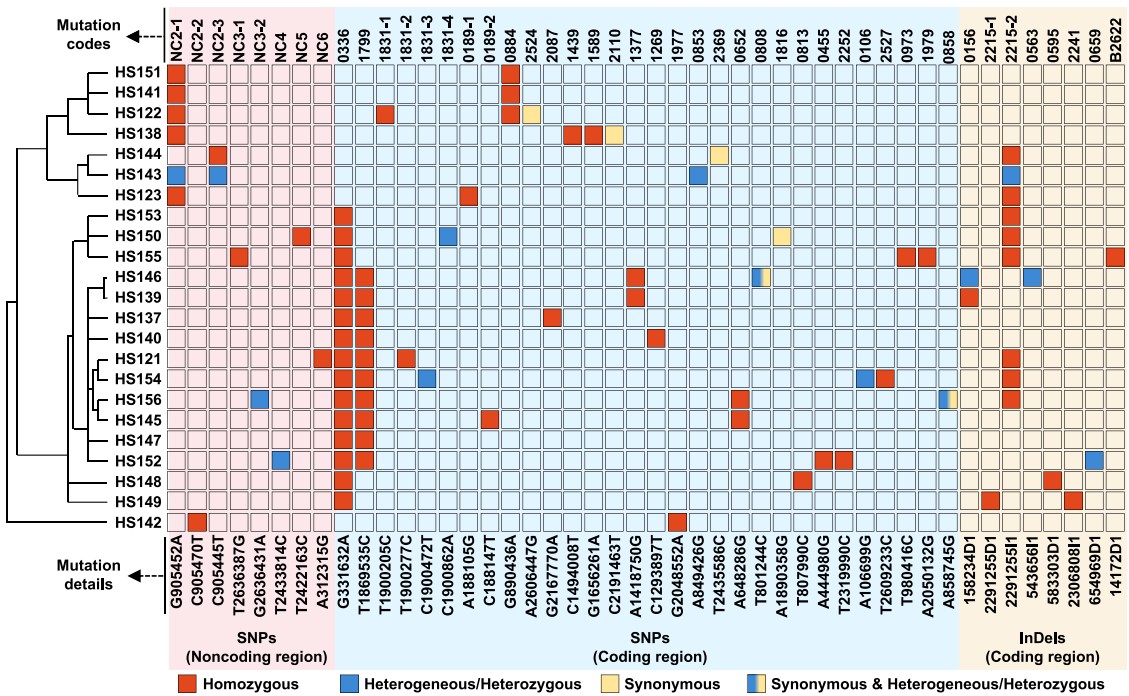

**Fig. 4 | Genomic mutations in 23 evolved *Synechococcus* strains with improved HLHT tolerance.** Hierarchical clustering analysis was performed for the 23 strains based on the similarities of identified mutations (left). According to the mutation loci and types, all the mutations were classified into three groups, namely the SNP group in the non-coding region (red background), the SNP group in the coding region (blue background), and the InDel group in the coding region (yellow background). The codes and details of the mutations are listed at the top and bottom, respectively. The mutation details represent the base mutation in the genome. For example, the SNP of G905452A represents the mutation of the 905452 from base G to A at the genome, and the InDel of 158234D1 refers to the deletion of the base 158234 from the genome. For ease of illustration, the mutations were coded according to the entry codes of the genes in KEGG. For example, as the G890436A mutation was located in the *synpcc7942_0884* gene and there were no other mutations identified in this gene, the mutation was termed 0884, whereas,

for multiple mutations located in the same regions or genes, additional codes were supplemented. For example, the four non-synonymous mutations (T1900205C, T1900277C, C1900472T, and C1900862A) located in *synpcc7942_1831* gene were termed as 1831-1, 1831-2, 1831-3, and 1831-4, respectively. Similarly, the five non-coding intergenic regions were coded as NC2-NC6, and the mutations located in such regions would be coded following the same principle of mutations found in coding regions. For example, the three mutations (G909452A, C905470T, and C905445T) located in the intergenic region between *synpcc7942_0894* and *synpcc7942_0895* were coded as NC2-1, NC2-2, and NC2-3. Mutations have been represented with different colors according to the sequencing results. Orange, homozygous; blue, heterogeneous/heterozygous; yellow, synonymous; half blue and half yellow: synonymous and heterogeneous/heterozygous. Source data are provided as a Source data file.

machinery in *Synechococcus* cells, guaranteeing the stability of phenotypes and genotypes (Supplementary Fig. 8).

Besides the application in direct screening mode under lethal conditions, the engineered hypermutation system of *Synechococcus* also showed advantages in accelerating the adaptive laboratory evolution process through passages in liquid culture. As shown in Supplementary Fig. 9, compared with WT control, cells from both the engineered hypermutation process and MMS mutagenesis treatment showed improved adaptation capacities to high temperature and high light stress (42 °C and 1500/2000 µmol photons/m²/s). And when lifting selective pressure (45 °C and 2500 µmol photons/m²/s), the engineered hypermutable cell groups showed advantages over the MMS treating group, indicating that stronger genetic and phenotypic diversities might be endowed by the higher genomic mutagenic strengths.

**Mining functional mutations from HLHT-tolerant *Synechococcus* strains**

To explore the HLHT tolerance mechanism of *Synechococcus* obtained in the engineered hypermutable process, 23 tolerant strains were selected for genome sequencing. A total of 88 mutations were identified from the strains (including HS121, HS122, and HS123), with an average of 3.8 mutations per strain (Fig. 4, Supplementary Tables 2–4, and Supplementary Data 1). The number of mutations in each strain ranged from two to six. Among them, a small fraction was

heterogeneous/heterozygous (*Synechococcus elongatus* is polyploid and carries more than one copy of a chromosome in each cell, and thus alleles carrying different SNPs could be harbored in the mutants; the heterogeneous/heterozygous status indicates that both the mutation and WT peaks could be identified from the sequencing reads). After excluding the mutations that repeatedly occurred in more than one strain, a total of 46 types of genetic mutations were identified from the evolved strains, including 38 SNPs and 8 InDels (three small insertions and five small deletions). All the InDels were located in the CDS regions, resulting in frameshifts of the encoded proteins. Of the 38 SNPs, 30 were located in the CDS regions (23 non-synonymous mutations, six synonymous mutations, and one nonsense mutation) and the other eight were located in five non-coding intergenic regions. For ease of illustration, the 46 mutations were coded based on their locations (KEGG entry codes). For example, the G890436A mutation is located in the *synpcc7942_0884* gene, and this mutation is denoted as 0884; whereas the four mutations (T1900205C, T1900277C, C1900472T, and C1900862A) located in *synpcc7942_1831* gene were denoted as 1831-1, 1831-2, 1831-3, and 1831-4, respectively. The five non-coding intergenic regions were denoted as NC2–NC6 and the mutations located in these regions were denoted following the same principle of mutations found in coding regions. For example, three mutations (G909452A, C905470T, and C905445T) located in the intergenic region between *synpcc7942_0894* and *synpcc7942_0895* were coded as NC2-1, NC2-2, and NC2-3, respectively.

Hierarchical clustering analysis was performed for the 23 evolved strains according to the carried mutations[30]. As shown in Fig. 4 (left), the grouping of the evolved strains was determined by the 0336 (AtpA-C252Y) and NC2 (including NC2-1, NC2-2, and NC2-3, upstream the shikimate kinase gene) mutations, and each of the 23 strains carried one of the two mutation types without any intersections. Thus, it can be assumed that these two mutations might perform essential functions for *Synechococcus* cells to adapt to HLHT stress. To screen for additional functional mechanisms related to HLHT tolerance, the relationships and specificities of the mutations and afflicted genes were further analyzed and sorted. Mutations occurring more frequently (in multiple strains) may be related to the tolerant phenotypes. Besides 0336 mutation (65.2%) and NC2 (34.8%), 1799 (L199P in the hydrogenase formation protein, HypE-L199P; 43%) and 2215 (InDels in the large subunit ribosomal protein L15 encoding gene *Synpcc7942_2215*; 39%) also occurred with relatively high frequencies, accompanying either 0336 or NC2 mutations. Further, for strains carrying only limited mutations, the mutations may have a higher correlation with phenotypic improvements. Among the 23 evolved strains, 5 carried only two mutations (HS151 and HS141: NC2-1 and 0884; HS142: NC2-2 and 1977; HS147: 0336 and 1799; HS153: 0336 and 2215-2). In addition to the effects of NC2 and 0336 mutations, the other mutations could also contribute to the adaptation process. For multiple mutations that occur in the same genes or regions as independent events, the related proteins could be promising targets for phenotype improvement. Thus, mutations in the NC2 region, NC3 region, *synpcc7942_1831*, and *synpcc7942_0189* could be potential targets for HLHT tolerance improvement.

Among the selected mutations and the located genes, some have been previously reported to be involved in the adaptation of cyanobacteria to HLHT stress, e.g., AtpA (C252Y) in *Synechococcus* cells[31], and NdhF (F124L) in *Synechocystis* sp. PCC 6803 (termed as *Synechocystis*)[19]. In addition, clear linkages could also be established between the functions of EF-Tu (*Synpcc7942_0884*), IMP dehydrogenase (*Synpcc7942_1831*), or GMP synthase (*Synpcc7942_0189*) and the underlying mechanisms of HLHT tolerance. Because the synthesis and repair of D1 protein are known to be essential for maintaining the activities of PSII in photosynthesis facing impairment of ROS, this process would cause increased consumption of GTP[10,32], for which the synthesis pathway involves the contribution of IMP dehydrogenase and GMP synthase. In addition, it was reported that EF-Tu is targeted by ROS[33] and that its overexpression could protect protein synthesis and PSII activities in *Synechocystis* cells subjected to strong illumination[34]. Thus, the 0884 mutation (EF-Tu-P164L) may influence cellular traits by utilizing similar mechanisms. To further confirm the functions of the identified mutations, especially those with uncharacterized mechanisms (e.g., NC2 mutations), the mutations were reconstituted in the WT strain for phenotype evaluation.

## Identifying the primary mutations endowing HLHT tolerance to *Synechococcus*

A previously developed rapid evaluation strategy integrating mutation-containing homologous recombination and survival selection was adopted to determine the effects of the isolated mutations on HLHT tolerances (Fig. 5a)[23]. During the assay, plasmids carrying fragments containing specific mutations (Supplementary Data 2) were transformed into *Synechococcus* cells for potential homologous recombination with the chromosomes, and the positive modifications imparting improved tolerance can be screened out by isolating the surviving host cells. Previously, the 0336 (AtpA-C252Y) was introduced into *Synechococcus* chromosome using this strategy; thus, it was used as the positive control in this experiment. Homologous fragments containing some of the other mutations (NC2-1, NC3-1, 0884, 1799, 1977, 1831-1, 0189-1, and 2215-2) were constructed to transform *Synechococcus* cells, followed by selection under lethal light intensities and

temperatures for WT. Among the evaluated mutations, only the NC2-1 mutation proved effective in improving the HLHT tolerance in *Synechococcus* (Fig. 5a and Supplementary Fig. 10). In further evaluations, the same effects were confirmed for NC2-2 and NC2-3, indicating that mutations in the NC2 region were highly correlated with the acquisition of HLHT tolerance. In addition, a reverse genetics strategy was also utilized to construct recombinant strains carrying the mutations for evaluation (Fig. 5b). Consistent with the results obtained from the survival selection strategy, only the introduction of NC2 mutations could facilitate the survival and growth of recombinant strains (HS190, G905452A, sp^R; HS191, C905470T, sp^R; and HS192, C905445T, sp^R) under lethal conditions (Fig. 5b and Supplementary Figs. 11 and 12), while the strains carrying other mutations (e.g., HS177, 1831-1, sp^R) showed no significant improvements compared to the respective controls (Fig. 5b). Combining the above phenotypic evaluation results of the recombinant strains with the previously obtained occurrence frequency pattern in the evolved strains, it became clear that NC2 (upstream non-coding sequence of *synpcc7942_0894*) and 0336 (AtpA-C252Y) mutations were essential to drive the adaptation process of *Synechococcus* cells under HLHT stress.

Although mutations of the other loci were ineffective in enhancing the HLHT tolerance of *Synechococcus* when introduced alone, they might still play some roles in synergy with the two primary factors. In recombinant strains carrying 1831-1 (HS228, in Fig. 5c) or 0189-1 (HS230, in Fig. 5d) mutation in addition to the NC2-1 mutation, cell growth under HLHT stress was further optimized, compared with the respective controls, while no differences were detected in the growth rates when the strains were cultivated under normal conditions (Supplementary Fig. 13). After complementation, the HS241 strain was obtained based on the evolved strain HS141 and utilized for comparison with the HS199 strain carrying only the NC2-1 mutation. Advantages could be detected due to the supplementation of the EF-Tu mutation in HS241 under HLHT stress (Fig. 5e) and no differences were detected in the growth rates under normal conditions (Supplementary Fig. 13). Similar effects were also observed when comparing the HL7942 strain carrying only the 0336 mutation and the HS247 strain with an additional 1799 mutation (Fig. 5f); however, the growth of the HS247 strain was inhibited compared with that of HL7942 at normal conditions (Supplementary Fig. 13), indicating that the growth advantages imparted by the 1799 mutation under the HLHT conditions were obtained by sacrificing photosynthesis efficiencies under normal physiological conditions.

## NC2 mutation up-regulated the shikimate kinase expression to enhance HLHT tolerance in *Synechococcus*

NC2 and 0336 were identified to be the two primary mutations endowing HLHT tolerance to *Synechococcus*. Since the functions and mechanism of the 0336 mutation have been previously deciphered[31], we focused on the effects and mechanisms of NC2 mutations in the following study. As shown in Fig. 6a, b, the NC2-1 mutation broadened the HLHT adaptation range of *Synechococcus* cells. The HS199 strain (carrying the NC2-1 mutation) maintained robust growth at temperatures ranging from 30 °C to 45 °C and light intensities up to 2500 μmol photons/m²/s, and the optimum growth was obtained at 42 °C and 1500 μmol photons/m²/s.

The NC2 mutations were located in the upstream non-coding region of *synpcc7942_0894* gene encoding shikimate kinase and the downstream region of the *synpcc7942_0895* gene encoding a hypothetical protein. We supposed that they might influence the HLHT tolerance of *Synechococcus* through the regulation expression of *synpcc7942_0894* and *synpcc7942_0895*. Our experiment showed that under normal conditions (NLNT; 30 °C and 500 μmol photons/m²/s; the same below), the transcription of *synpcc7942_0894* gene was elevated by 17.4-fold in HS199 compared to that in the WT, while under HLHT conditions (HLHT; 42 °C and 2000 μmol photons/m²/s; the same

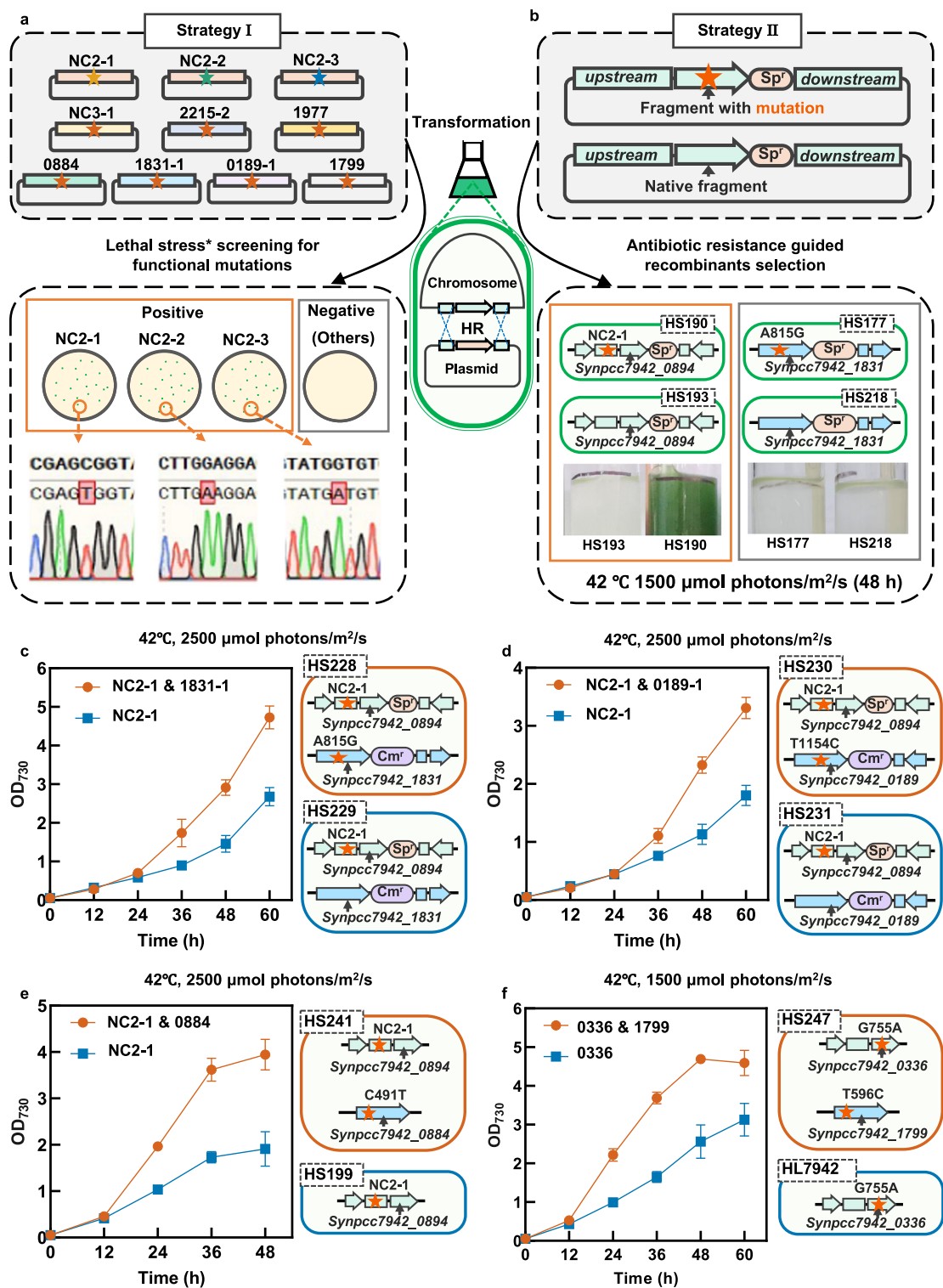

below), it increased 25-fold higher than that in the WT at NLNT (Fig. 6c). In comparison, transcription of *synpcc7942_0895* only increased by 1.7-fold and 2.5-fold in HS199 under NLNT and HLHT conditions, respectively. Consistent with the elevated transcription levels, western blot analysis also confirmed that the abundance of the *synpcc7942_0894* encoded proteins (shikimate kinase) was improved in the HS199 strain compared with that in WT and another HLHT tolerant strain HL7942 carrying the 0336 (AtpA-C252Y) mutation (Fig. 6d and Supplementary Fig. 14). Thus, it could be assumed that the enhanced tolerance to HLHT stress endowed by the NC2 mutation might be facilitated

through the elevated expression of shikimate kinase. To confirm this hypothesis, recombinant strains overexpressing *synpcc7942_0894* and *synpcc7942_0895* were constructed and evaluated for HLHT tolerance. As shown in Fig. 6e, f, *synpcc7942_0894* overexpression led to growth capacities of the recombinant *Synechococcus* strain similar to that of the HS199 strain carrying the NC2-1 mutation, whereas overexpression of *synpcc7942_0895* failed to improve the survival of the recombinant strain. The shikimate pathway is widely present in bacteria, fungi, algae, and plants and plays an essential role in the synthesis of aromatic amino acids and a large number of downstream metabolites[35–37].

**Fig. 5 | Identification of the functional mutations contributing to the improved HLHT tolerance in *Synechococcus* cells. a, b** Illustrations of the two strategies utilized for evaluating the effects of the putative functional mutations on HLHT tolerance improvement. In strategy I, homologous fragments containing mutations (NC2-1, NC2-2, NC2-3, NC3-1, 0884, 1799, 1977, 1831-1, 0189-1, and 2215-2) were assembled in plasmids and transformed into *Synechococcus* cells, which would be screened under lethal light intensity and temperature of the WT control. Survival colonies could only be obtained from the transformed manipulations with mutations of NC2-1, NC2-2, or NC2-3 (**a**). For strategy II, the mutations were introduced into *Synechococcus* chromosomes through antibiotics-resistance-based selection. Meanwhile, recombinant strains carrying only the antibiotic resistance gene at the same locations (without the respective mutations) were constructed as controls. The parallel constructed strains (with or without a specific mutation) would be cultivated and compared in HLHT conditions (42 °C and 1500 μmol photons/m²/s) to evaluate the effects of the specific mutation. The comparisons between HS193 versus HS190 (with or without the NC2-1 mutation) and HS177 versus HS218 (with or without the 1831-1 mutation) are shown as examples (**b**). **c** The synergistic effects of the 1831-1 mutation and the NC2-1 mutation (HS228) on improving cell growths of the strain only carrying NC2-1 (HS229) at the conditions of 42 °C and 2500 μmol photons/m²/s. **d** The synergistic effects of the 0189-1 mutation and the NC2-1 mutation (HS230) on improving cell growths of the strain only carrying NC2-1 (HS231) at the conditions of 42 °C and 2500 μmol photons/m²/s. **e** The synergistic effects of the 0884 mutation and the NC2-1 mutation (HS241) on improving cell growths of the strain only carrying NC2-1 (HS199) at the conditions of 42 °C and 2500 μmol photons/m²/s. **f** The synergistic effects of the 1799 mutation and 0336 mutation (HS247) on improving cell growths of the strain only carrying 0336 mutation (HL7942) at the conditions of 42 °C and 1500 μmol photons/m²/s. Each experiment was replicated more than twice to ensure its reliability. Data are presented as mean values ± SD (*n* = 3 biological replicates). Source data are provided as a Source data file.

Additionally, shikimate kinase (EC:2.7.1.71) is an important enzyme in the shikimate pathway that catalyzes the phosphorylation of shikimate to shikimate-3-phosphate. To explore whether the enhancement of shikimate kinase expression could work as an effective strategy for photosynthesis optimization in other cyanobacterial species, the homolog of *synpcc7942_0894* (*sll1669*) was overexpressed in *Synechocystis*. At 30 °C and 38 °C, the recombinant *Synechocystis* strain overexpressing the *sll1669* gene showed optimized adaptation to increased light intensities (Fig. 6g), indicating that the overexpression of shikimate kinase might be a universal strategy for adapting cyanobacterial photosynthesis to light and temperature stress.

## NC2 mutation remodeled the photosynthetic metabolism in *Synechococcus*

For more comprehensive clarifications of the NC2-1 mutation effects, global RNA-sequencing analysis was performed on the HS199 strain (NCBI Sequence Read Archive, PRJNA847037). Meanwhile, the transcriptional abundance of the key genes mentioned below was verified by qPCR (Supplementary Fig. 15), and the protein abundance changes were determined by Parallel Reaction Monitoring (PRM). At the NLNT condition, 14% of the transcripts in HS199 were up-regulated and 15% were down-regulated (Supplementary Fig. 16 and Supplementary Data 3). Among the upregulated transcripts, *synpcc7942_0894* showed a 20-fold increase (Supplementary Fig. 17), which was consistent with the results obtained by qPCR. Enrichment analysis revealed that the oxidative phosphorylation pathway was upregulated (*p* = 0.01) in HS199 (Supplementary Fig. 18). A large portion of the genes, such as genes encoding NAD(P)H dehydrogenase (NDH), cytochrome c oxidase, succinate dehydrogenase, and FoF1-ATP synthase, as well as the genes contributing to phosphate metabolism, were upregulated. Similar changes were also found in the PRM results (Supplementary Data 4), in which the protein abundances of 4 subunits in NDH complex and 6 subunits in FoF1-ATP synthase were increased by 2 to 4 folds. In addition, several genes participating in energy metabolism during photosynthesis also showed differential expression in HS199. For example, genes encoding ferredoxin (*synpcc7942_1499*), the three subunits of H⁺ translocating NAD(P) transhydrogenase (*synpcc7942_1610, synpcc7942_1611, synpcc7942_1612*), and the HoxU (*synpcc7942_2557*) subunit of the hydrogenase complex were all upregulated. The protein abundance of the synpcc7942_1612 (β subunit of the H⁺ translocating NAD(P) transhydrogenase) was found 30-fold increased than that of WT in the PRM assay. Although transcriptions of some components in PSII and PSI complexes were found downregulated (including D1 and PsaB), the PRM assay revealed no significant changes in protein abundances of PsaA, PsaB, PsaC, and PsaD in PSI, and even an increased (2.86-fold) abundance of D1 (PsbA1, *Synpcc7942_0424*) in PSII. It could be speculated that the photosynthetic chain in *Synechococcus* was remodeled by the NC2-1 mutation to enhance oxidative phosphorylation and cyclic electron transfer activities for adequate ATP synthesis (Fig. 7), which could guarantee accelerated cell growth in HS199. For verification, we compared the cyclic electron flow (CEF) of HS199 and WT, by monitoring the kinetics of P700⁺ re-reduction and the redox kinetics of P700. As shown in Fig. 8, the initial rate of P700⁺ re-reduction in HS199 was higher than that of WT (*p* = 0.0042), while the P700 re-oxidation was slower in HS199, indicating that the CEF activity of HS199 was enhanced. Besides, increased respiration rate and whole chain O₂ evolution rate were also observed in HS199 in both the NLNT and HLHT conditions, which was consistent with accelerated cellular growth and biomass production. In addition to the photosynthesis system, the two-component system pathways, including components involved in the activities of rhythm signal transduction, phototaxis regulation, bacterial movement, and nitrogen uptake, were downregulated (*p* = 0.0019) (Supplementary Fig. 19), which might allow *Synechococcus* cells to conserve resources from hierarchical responsive and regulatory systems under stable cultivation environments.

Most of the above-mentioned changes induced by NC2 mutation in *Synechococcus* transcriptome were further enhanced in the same direction when cells were cultivated under HLHT conditions. 10% of the genes in HS199 were upregulated, whereas 8% of the genes were downregulated at HLHT than that at NLNT (Supplementary Data 5), and among the upregulated transcripts, the oxidative phosphorylation pathway was still overrepresented (*p* = 0.0169) (Supplementary Figs. 16 and 18). Transcripts of the three H⁺ translocating NAD(P) transhydrogenase subunits were further elevated 40-fold higher than that in the WT under NLNT conditions, indicating that conversion between NADH and NADPH might play an important role in regulating redox and energy balance of electron carriers in photosynthesis under HLHT conditions. The two-component system pathway (*p* = 0.006) was also further downregulated in HS199 under HLHT compared to the NLNT conditions (Supplementary Fig. 19). Coping with the changes in photosystem and energy metabolism in the HS199 strain under HLHT conditions, the carbon fixation and metabolism network were also regulated to adapt the changed supply of ATP and NADPH. The transcriptions of several important enzymes contributing to carbon fixation, including phosphoglycerate kinase (EC:2.7.2.3), fructose-1,6-bisphosphatase I (EC:3.1.3.11), fructose-6-phosphate phosphoketolase (EC:4.1.2.9), and glyceraldehyde 3-phosphate dehydrogenase (EC:1.2.1.12), were increased in HS199 under HLHT compared to that in the WT under NLNT conditions and could be related to the enhanced biomass production performance of HS199 under HLHT conditions (Supplementary Fig. 20). In addition, many other transcripts involved in other carbon metabolism pathways, including glycogen metabolism, the Embden–Meyerhof–Parnas pathway (glycolysis, EMP), pentose phosphate pathway (PPP), and tricarboxylic acid cycle (TCA), were also affected, indicating comprehensive rearrangement of the carbon metabolism network (Supplementary Fig. 21). To explore the respective effects on cellular metabolism and physiology, biomass and

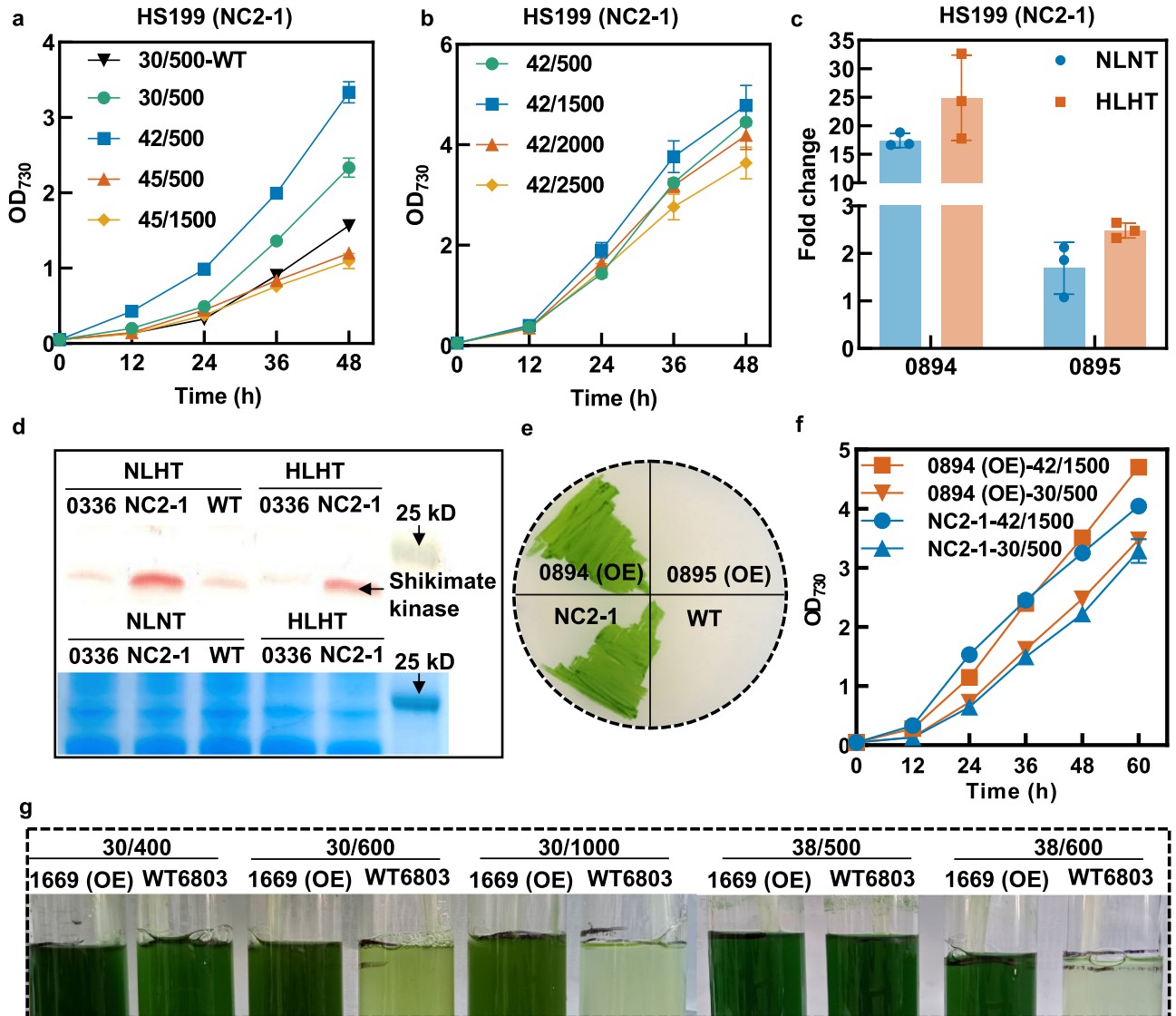

**Fig. 6 | NC2 mutation enhanced the expression of shikimate kinase to improve HLHT tolerance in *Synechococcus*. a** Cell growths of the strain HS199 (carrying the NC2-1 mutation) when cultivated at increasing temperatures (from 30 °C to 45 °C) and light intensities (from 500 to 1500 μmol photons/m²/s). **b** Cell growths of the strain HS199 when cultivated at 42 °C with increasing light intensities (from 500 to 2500 μmol photons/m²/s). **c** Relative mRNA abundances of the genes flanking the NC2-1 mutation (upstream: *synpcc7942_0895* gene, 0895; downstream: *synpcc7942_0894* gene, 0894) in HS199 strain under normal conditions (NLNT; 30 °C and 500 μmol photons/m²/s; the same below) and HLHT conditions (HLHT; 42 °C and 2000 μmol photons/m²/s; the same below). The mRNA abundances were normalized relative to that of the corresponding gene in WT cells under NLNT conditions to determine the fold changes. **d** Western blot analysis (up) and SDS-PAGE (down) of the *synpcc7942_0894* encoded protein, shikimate kinase in HS199 strain. HL7942 strain carrying the 0336 mutation and showing HLHT tolerance was utilized as an additional control besides the WT to evaluate the effects of

HLHT cultivation. **e** Effects of overexpression of *synpcc7942_0894* [0894 (OE)] and *synpcc7942_0895* [0895 (OE)] in *Synechococcus* for survival and growth on BG11 agar plates cultivated in the incubator at 44 °C and 500 μmol photons/m²/s (warm white light). HS199 and WT strains were used as positive and negative controls, respectively. **f** The effects of overexpression of *synpcc7942_0894* gene [0894 (OE)] in *Synechococcus* on cell growth under NLNT and HLHT conditions. The HS199 strain was used as a control. **g** The effects of shikimate kinase gene (*sll1669*) over-expression in *Synechocystis* on cell growth under increased temperatures (from 30 °C to 38 °C) and light intensities (from 400 to 1000 μmol photons/m²/s). The recombinant strain was termed 1669 (OE), and the wildtype *Synechocystis* sp. PCC 6803 (WT6803) was used as the control. The cultures were photographed after 48 h. Each experiment was replicated more than twice to ensure its reliability, including **d**. Data are presented as mean values ± SD (*n* = 3 biological replicates). Source data are provided as a Source data file.

intracellular glycogen contents of HS199 and WT cells cultivated under NLNT and HLHT conditions were determined. As shown in Supplementary Fig. 22, accompanied by the enhanced biomass accumulation in total and on per cell levels, HS199 cells possessed nearly threefold and ninefold higher glycogen contents than that of WT under NLNT and HLHT conditions, respectively, which might serve as an important sink mechanism to accept the enlarged carbon output from enhanced photosynthesis activities and the increased ATP supply from the activated CEF pathway.

Another noteworthy change in HS199 photosynthesis was the reduction of PSII activities. In NLNT conditions when HS199 possessed enhanced overall photosynthesis activities and cell growth, PSII-mediated $O_2$ evolution rate of HS199 was lower than that of WT (Supplementary Fig. 23a). Additionally, with gradually increased light intensities during detection, PSII activity of HS199 (peaking at 0.02 mg $O_2 L^{-1} OD_{730}^{-1} s^{-1}$) remained lower than that of WT (peaking at approximately 0.025 mg $O_2 L^{-1} OD_{730}^{-1} s^{-1}$). When treated with HLHT stress (42 °C and 2500 μmol photons/m²/s), the PSII activities of both

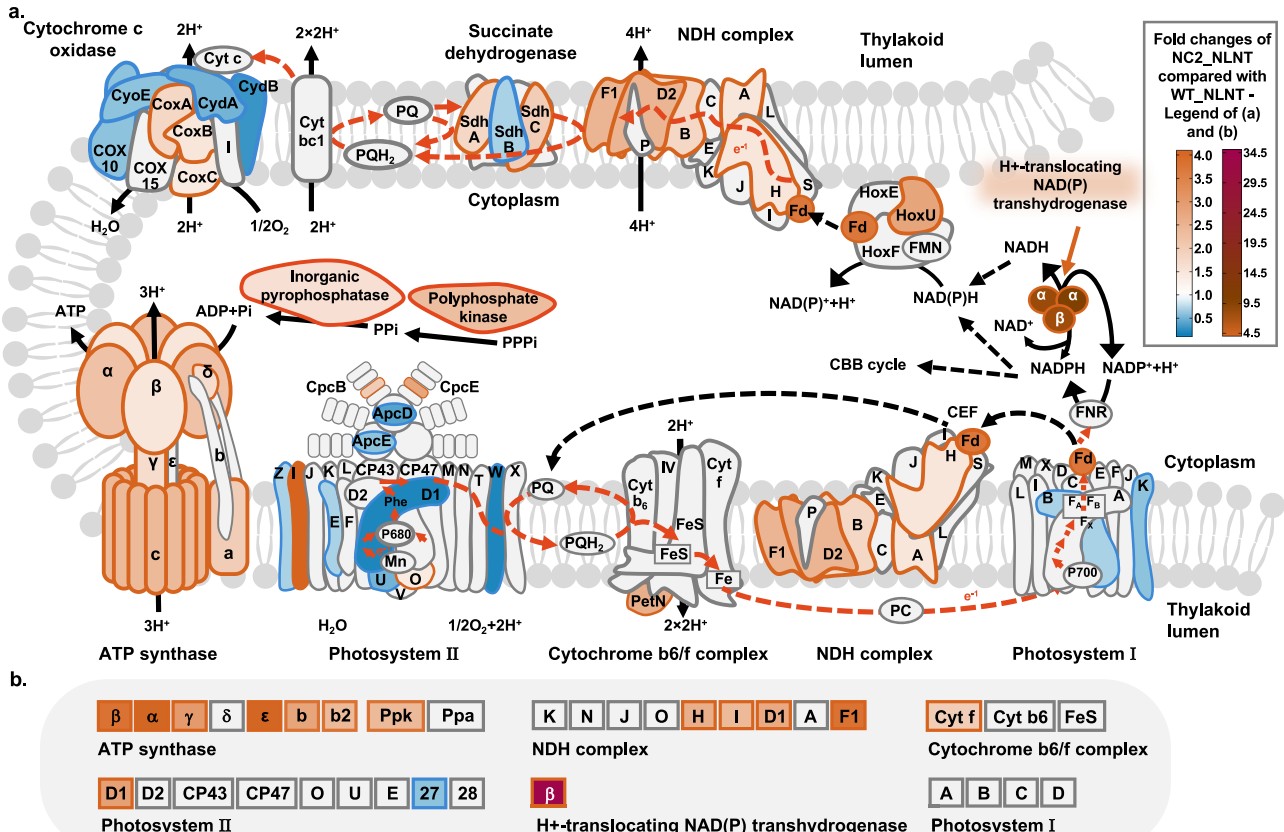

**Fig. 7 | Changes of photosynthetic chain and oxidative phosphorylation pathway caused by the NC2-1 mutation in *Synechococcus*. a** Transcriptions of the genes involved in the photosynthetic chain and oxidative phosphorylation pathway were compared between HS199 and WT. **b** Protein abundances of important components for photosynthesis and oxidative phosphorylation of HS199 and WT were calculated and compared. WT and HS199 strain were cultivated and sampled under normal conditions (30 °C and 500 μmol photons/m²/s) (*n* = 3 biological replicates). The transcriptomes were sequenced using RNA-seq and the protein content was determined using PRM. The components and structures of the photosynthetic chain and oxidative phosphorylation pathway were illustrated by referring to the KEGG (syf00190 and syf00195) and relative publications[72–75], and some modifications were made. For each complex, the annotation of the single subunit is abbreviated. For example, proteins in photosystem II, such as PsbZ, PsbI, and PsbJ, are labeled as Z, I, and J, respectively. Besides, Ppa refers to polyphosphate kinase and Ppk refers to inorganic pyrophosphatase. The fold changes of specific genes in HS199 compared with those in WT are represented by the colors of the respective proteins. Source data are provided as a Source data file.

WT and HS199 cells would be reduced (*p* < 0.0001 for WT and *p* = 0.0098 for HS199 after 30 min of exposure to HLHT stress) (Supplementary Fig. 23b, c), which is in consistence with the downregulated photosystem pathway from RNA-seq analysis (*p* = 0.0004 for HS199 and *p* = 0.0007 for WT). Meanwhile, among the 9 subunits of PSII detected in PRM analysis, abundances of 5 subunits were reduced in HS199 cultured under HLHT, while in WT cells with HLHT treatment, 2 subunits were down-regulated and one was up-regulated; and correspondingly, PSII activities of HS199 were also lower than that of WT during the HLHT treatment. When lincomycin was supplemented to WT and HS199 cells exposed to HLHT stress, the PSII activities of both strains would be decreased (to 19.4% in HS199 and to 14.8% in WT of the initial levels in 30 min, Supplementary Fig. 23c), and slight advantages of H199 in PSII stability facing HLHT stress might be related to the high abundance of D1 protein (Supplementary Data 4). In addition, HS199 exhibited higher $F_v/F_m$ than WT under NLNT and after being exposed to HLHT, the $F_v/F_m$ of WT and HS199 both decreased while HS199 showed lower $F_v/F_m$ than WT (Supplementary Fig. 24). The above results suggested that HS199 might adopt the strategy to weaken PSII activity for reducing the overall light energy harvesting and avoiding over-reduction stress, and meanwhile to enhance energy transfer and utilization efficiency (through enhanced CEF and respiration), finally to adapt the HLHT environment and realize rapid growth.

When comparing the transcriptome of HS199 under HLHT with that of the WT under NLNT conditions, pathways for ribosome synthesis (*p* = 0.0001) would also be overrepresented from the upregulated transcripts (Supplementary Fig. 18 and Supplementary Data 6), which might be related to cellular tolerance under HLHT conditions, and the whole metabolic network for protein synthesis might be regulated during the adaptation process of *Synechococcus* cells. As mentioned above, the ribosome synthesis pathway was enhanced in HS199 cells under HLHT (26 out of 53 related transcripts were upregulated, Fig. 9a); meanwhile, 10 out of 25 genes participating in the aminoacyl-tRNA biosynthesis pathway were also upregulated (Fig. 9b). Moreover, the transcription of translation elongation factors, including EF-G, EF-G2, EF-Tu, and EF-Ts, was also upregulated (Fig. 9c). The transcription of genes encoding ferredoxin-nitrite reductase (EC:1.7.7.1) and glutamine synthetase (EC:6.3.1.2), involved in amino acid metabolism, was also upregulated (Supplementary Data 6). Considering the above changes, it can be speculated that the protein synthesis capacity of HS199 cells might be enhanced under HLHT conditions. This hypothesis can also be corroborated by the physiological data since the intracellular protein content in HS199 under the corresponding conditions was higher than that in the WT (Fig. 9d). Regarding photosynthetic activities under HLHT stress, enhanced protein synthesis capacities might guarantee the renewal of the impaired proteins and maintenance of biomass production.

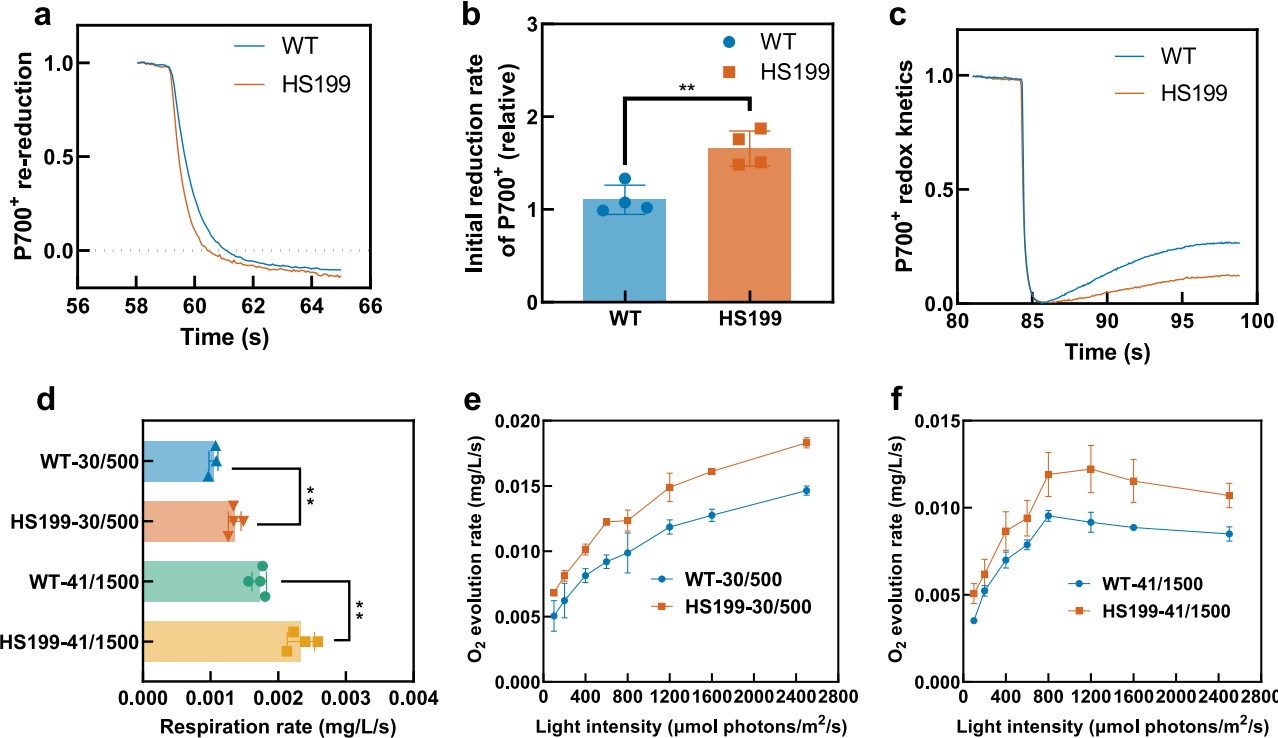

**Fig. 8 | Photosynthetic physiological parameters of WT and HS199. a–c** WT and HS199 cells were suspended in BG11 containing 20 µM DCMU and dark-acclimated for 20 min before measurement. The P700 was oxidized by FR light for 40 s, and the subsequent re-reduction of P700+ in the dark was monitored. The curves were normalized by equating the absorbance maximum of 1 s before termination of FR light to one (**a**). The initial rate of P700+ re-reduction was acquired from normalized curves by calculating the slope of the initial values after FR was turned off and the $R^2$ was more than 0.99 (**b**). P700 redox was monitored after the termination of AL illumination under a background of FR light. The P700 levels were normalized by equating the absorbance minimum after the termination of AL illumination to 0 and equating the absorbance maximum 3 s before termination of AL illumination to one (**c**). **d** The respiration rate of WT and HS199 under 30 °C and 500 µmol

photons/m²/s (30/500) and 41 °C and 1500 µmol photons/m²/s (41/1500). (Oxygen consumption per second by 1 L cells with 1 OD730 in dark). **e, f** Whole-chain O2 evolution of WT and HS199 under 30 °C and 500 µmol photons/m²/s (30/500) (**e**) and 41 °C and 1500 µmol photons/m²/s (41/1500) (**f**). (Total oxygen evolution per second by 1 L cells with 1 OD730 under different light conditions). Each experiment was replicated more than twice to ensure its reliability. Data in **a**, **c** are presented as mean values of 4 biological replicates, and data in **b–e** are presented as mean values ± SD. The sample size *n* (biological replicates) is 4, 4, 3, and 3 in **b**, **d–f** except for that *n* of WT-30/500 in **d** is 3. Statistical analysis was performed using a two-tailed unpaired Student's *t* test (**p < 0.01). The *p* value is 0.0042 in **b**. The *p* values are 0.0045 and 0.0017 in **d** (from up to down). Source data are provided as a Source data file.

Some additional protective mechanisms might also be adopted by *Synechococcus* cells carrying the NC2 mutation to adapt to the HLHT conditions. For example, the transcription of the photosynthesis-antenna protein pathway ($p = 6.5 \times 10^{-5}$), as well as the photosynthesis ($p = 0.0004$) in HS199 was significantly downregulated under HLHT compared to the NLNT conditions, which was also observed for WT under HLHT and NLNT conditions (Supplementary Fig. 18 and Supplementary Data 5 and 7). And the amount of chlorophyll *a* in HS199 under HLHT conditions was decreased to approximately 39% of WT under NLNT conditions (Supplementary Fig. 25). The changes were believed to be effective in reducing excessive energy absorption and avoiding photoinhibition. In addition, the scavenging of ROS by antioxidants, such as GSH, might also be enhanced as an effective photoprotection strategy. Compared with WT under NLNT conditions, transcripts for glutathione peroxidase (EC:1.11.1.9; catalyzing the hydrogen peroxide scavenging reactions converting GSH to GSSG) were upregulated by 4.8-fold. Simultaneously, glucose-6-phosphate 1-dehydrogenase (EC:1.1.1.49) and 6-phosphogluconate dehydrogenase (EC:1.1.1.44) in the pentose phosphate pathway (PPP) were upregulated by 9.7- and 6.4-fold, respectively, which could provide sufficient reducing power (NADPH) for the regeneration of GSH (Supplementary Fig. 26).

The global remodeling of the *Synechococcus* transcriptome carrying the NC2 mutation may also be caused by disturbances in the transcriptional regulatory hierarchy. The transcription of RpaA

(*Synpcc7942_0095*), a global rhythm output regulator, was downregulated in HS199 than that in WT under both HLHT and NLNT conditions (Supplementary Data 3, 5, and 6), and the downregulation of RpaA has been shown to be related to enhanced cell growth and HLHT tolerance of the fast-growing *Synechococcus* UTEX 2973 strain[31]. In addition, differential transcription of some other transcription factors was also detected in HS199, indicating comprehensive rewiring of the transcriptional regulation hierarchy (Supplementary Table 5 and Supplementary Data 3, 5, and 6).

## Discussion

Improving tolerances of cyanobacterial photosynthesis to abiotic stress is challenging due to the lack of effective approaches and clear underlying mechanisms. Long-term continuous cultivation and passages are usually required in the typical ALE process to accumulate positive functional mutations and to isolate the desired phenotypes[38–40]. To eliminate the restrictions of cyanobacterial evolution, we aimed to disrupt the genome replication fidelity machinery in cyanobacteria cells to accelerate the diversification process of offspring populations. Through manipulating the putative fidelity-related genetic elements, mutation rates of *Synechococcus* were increased by over 100-fold; and further improvement of in vivo mutagenesis strengths was obtained based on the synergy effects of the fidelity machinery defects with inhibitory light intensities and temperatures, resulting in 1000-fold increased mutation rates. This engineered

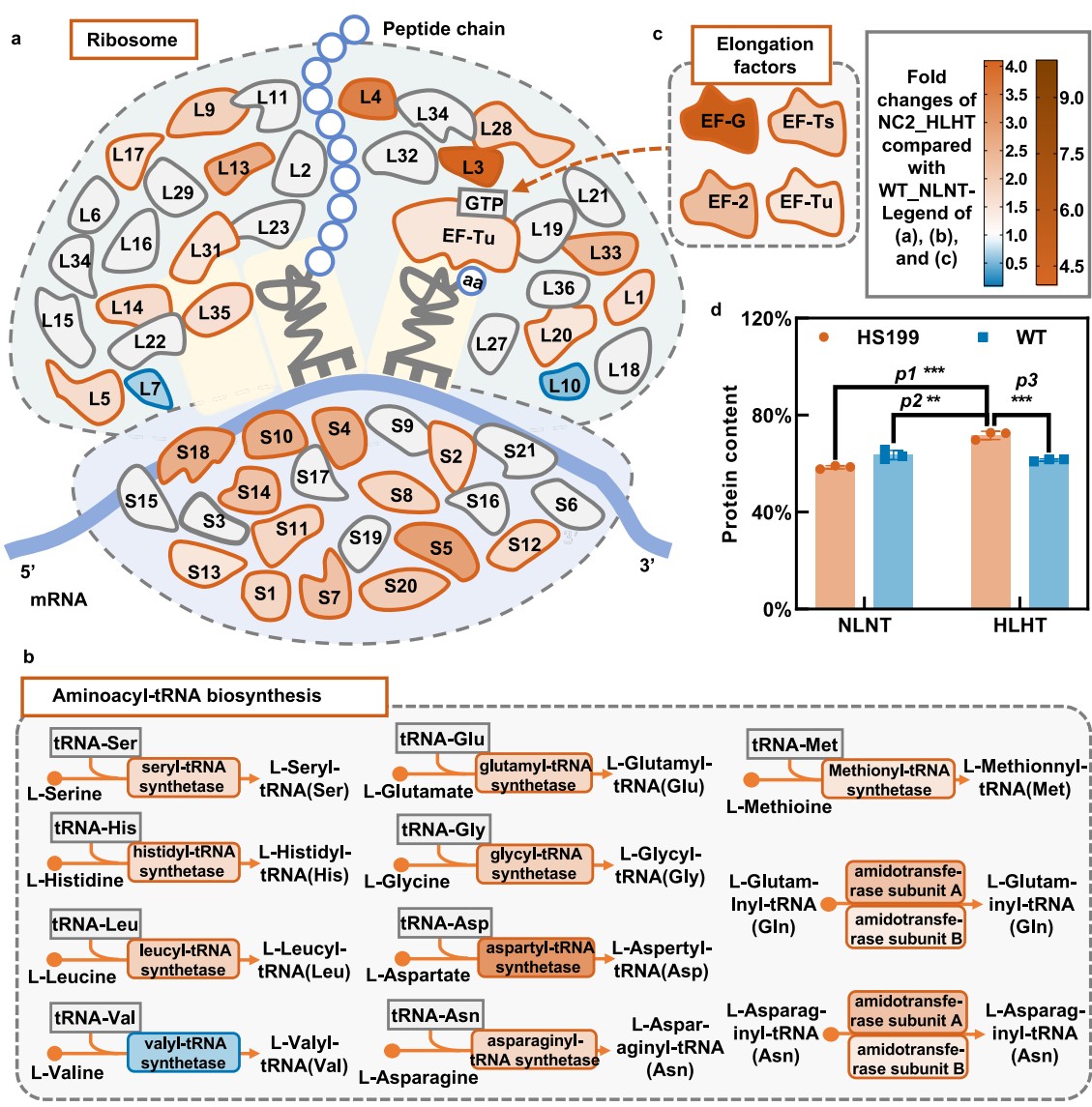

**Fig. 9 | Changes in transcript abundance of the genes participating in translation machinery of HS199 under HLHT compared to WT under NLNT conditions. a** Changes in transcript abundance of the genes encoding ribosome protein in HS199 strain. **b** Changes in transcript abundance of the genes participating in the aminoacyl-tRNA biosynthesis. **c** Changes in transcript abundance of the four genes encoding translation elongation factors. The colors filled in specific proteins represent the transcription abundance-fold changes of the respective genes in HS199 cells cultivated under HLHT compared with those in WT cells cultivated under NLNT conditions. **d** The protein contents of HS199 and WT cells cultivated under NLNT and HLHT conditions. The transcriptome of the HS199 strain cultivated under HLHT conditions (42 °C and 2000 μmol photons/m²/s) was sampled and sequenced. Data are presented as mean values ± SD ($n = 3$ biological replicates). Statistical analysis was performed using a two-tailed unpaired Student's $t$-test (**$p < 0.01$, ***$p < 0.001$). The $p$ values in **d** are 0.0002 (p1), 0.0052 (p2), 0.0005 (p3). Source data are provided as a Source data file.

hypermutation process mimicked the well-known stress-induced-mutagenesis (SIM) phenomenon, which was triggered by environmental stresses and played an important role in bacteria and yeast acquiring fitness and adaptabilities[41,42]. In the natural SIM process, multiple response mechanisms leading to an error-prone genome replication process are induced by abiotic stresses to increase genetic diversity and relieve selective pressure at the population level in a short period. In this study, the error-prone replication in *Synechococcus* caused by inhibitory illumination and temperature were amplified by artificial manipulations of the genes *mutS* and *recA*, the alleles that naturally participated in the typical SIM model[41,42]. Such an engineered hypermutation system could be expected to boost the acquisition of *Synechococcus* mutants and may also inspire the development of adaptive laboratory evolution approaches for other microorganisms.

The isolated HLHT-tolerant *Synechococcus* mutants permitted deciphering of the underlying mechanism through simple comparative

genomic analysis. It was revealed that the enhanced HLHT tolerance could be endowed by quite limited genome mutations (an average of 3.8 mutations per strain), which, together with some recent findings[19,31], show the great plasticity of the cyanobacterial genome. Loci analysis of the mutations revealed that some of the relevant functions could be directly associated with photosynthesis or some typical photoprotective mechanisms. For example, the 0336 (AtpA-C252Y) mutation has been proven effective for improving FoF1-ATP synthase activity in *Synechococcus*, and thus adapts to the enhanced photosynthetic electron transfer chain (ETC)[31]. The enhanced activity of FoF1-ATP synthase was important for relieving the stress from excess photosynthetic electrons and providing abundant energy to support the PSII repair facing photooxidative damage. Being the primary target in PSII damage, the de novo synthesis and repair of D1 protein are guaranteed by an abundant supply of ATP, as well as the derived synthesis of GTP. Genome mutations in two genes involved in the de novo synthesis of

GTP (*synpcc7942_1831* and *synpcc7942_0189*) might be beneficial for stable GTP supply under HLHT stress. GTP consumption is high during protein synthesis, and the abundant GTP supply could promote the regeneration of inactivated D1[43] and accelerate de novo synthesis[32]. Furthermore, an additional mutation was found in the EF-Tu protein (*synpcc7942_0884*), which plays an essential role in translation machinery, bringing aminoacyl-transfer RNA to the ribosome, and is closely coupled to GTP hydrolysis[44]. Strong illumination usually causes the oxidative damage of EF-Tu thus inhibiting PSII repair efficiencies[33], and the light-inducible elevation of EF-Tu expression is found to be closely correlated with enhanced PSII repair in *Synechocystis*[34]. The overexpression or modification (to rescue the oxidative damage sites) of EF-Tu could enhance protein synthesis and PSII repair while failing to restore growth inhibition[33,34]. The EF-Tu-P164L mutation discovered in this work optimized the HLHT tolerance of *Synechococcus* together with NC2-1 mutation, which could be expected to promote a detailed understanding of the oxidative protective mechanisms in cyanobacteria. In addition to the ATP and protein synthesis pathways, some functional mutations have been identified in genes related to NAD(P)H metabolism, such as the NAD(P)H dehydrogenase complex and [NiFe]-hydrogenase, which might contribute to enhancing and stabilize the electron transfer machineries as protective mechanisms against HLHT stress[19,45,46].

Among all the mutations, 0336 (AtpA-C252Y) and NC2 are identified to be the primary factors, effectively improving cellular HLHT tolerance individually. Unlike the structure-activity regulation effects of C252Y on FoF1-ATP synthase, the NC2 mutations influenced cellular properties by enhancing the shikimate kinase expression. Shikimate kinase catalyzes the conversion of shikimate to 3-phosphoshikimate, which is essential for the synthesis of aromatic amino acids, tryptophan, phenylalanine, tyrosine, and diverse derived metabolites. Based on the improved efficiency and stability of photosynthesis due to the overexpression of shikimate kinase in *Synechococcus* and *Synechocystis*, it could be speculated that the flux of the shikimate pathway might serve as the rate-limiting pathway for photosynthetic activities in cyanobacteria. The enhanced shikimate kinase expression might cause global impacts by synthesizing more regulatory metabolites. In higher plants and eukaryotic microalgae, the shikimate pathway provides precursors for the synthesis of important plant hormones such as salicylic acid and indole-3-acetic acid. Although functional enzymes have not been identified, the synthesis of such hormones has been detected during the cultivation of other *Synechococcus* species[47]. The potential synthesis of plant hormones and other secondary metabolites could give possible explanations about the transcriptome shifting in the HS199 strain carrying the NC2 mutation leading to the remodeling of the photosynthetic chain and enhanced oxidative phosphorylation activities, e.g., the upregulation of components in FoF1-ATP synthase (subunit alpha, beta, gamma, delta, a, and c) and NDH complex. The enhancement of shikimate pathway could also regulate the photosynthesis activities by influencing the synthesis of plastoquinone (PQ). PQ in cyanobacteria is synthesized utilizing chorismate, the end metabolite of the shikimate pathway, as important precursors[48], and widely participates in photosynthetic and respiratory electron transport chain, as well as in many other metabolic processes[49–51]. In energy metabolism, PQ works as electron carriers, shared by linear electron flow, cyclic electron flow, and other alternative electron flows[51]. The availability of PQ has been found to play an important role in photosynthesis acclimation to high-light and high-temperature conditions[52,53]. The up-regulation of the shikimate pathway in the strain carrying NC2 mutation could be expected to enhance PQ synthesis, and support more robust electron transport in HLHT conditions. On a physiological level, the most determining impact of NC2 mutation was the enhanced cyclic electron transport activities, which might be the synergistic results from enhanced PQ supply[54,55], up-regulated NDH complex, and reduced PSII activities[56,57]. CEF plays

an essential role in balancing the ATP/NADPH ratio (promoting the supply of additional ATP required by the high-energy consuming process, e.g., protein synthesis and materials transport) and preventing over-reduction damage of PSI and PSII in plants and algae for both short-term acclimation and long-term adaptation[58–60]. Previously, changes in shikimate pathway genes and related metabolites were observed in photoautotrophs with regulated CEF activities. In an *Arabidopsis* mutant with impaired CEF, levels of shikimate metabolites were significantly reduced[61], while in a *Chlamydomonas* strain exhibiting high CEF, two key enzymes in the shikimate pathway were up-regulated[60]. However, the interactions between the shikimate pathway and the CEF as well as other molecular mechanisms for stress response are unclear. Our work indicates that the enhanced CEF and the optimized photosynthesis stabilities facing HLHT stress in *Synechococcus*, could be a result of the up-regulation of the shikimate pathway. Another possible mechanism of the enhanced shikimate pathway to influence cyanobacterial photosynthesis activities is through directly enhancing aromatic amino acids supply for the requirement of photosystems synthesis. The proportion of diverse aromatic amino acids in 2657 proteins in *Synechococcus* was counted and classified according to the KEGG pathway and solubilities of the proteins (Supplementary Figs. 27 and 28 and Supplementary Data 8–11). As shown in Supplementary Fig. 28, the total proportions of the shikimate pathway-derived aromatic amino acids (tyrosine, phenylalanine, and tryptophan) in all proteins, soluble proteins, and membrane proteins were 7.7%, 7.3%, and 9.3%, respectively. The proportion in the photosynthesis pathway was the highest (10.2%), especially in photosystem II (14.5%), among all the annotated pathways (Supplementary Figs. 24 and 25 and Supplementary Data 7). Thus, it could be speculated that an enhanced supply of aromatic amino acids might guarantee the synthesis of essential photosystem components. Some other protective mechanisms are also pre-activated by NC2 mutation in *Synechococcus*. Previously, a mutated *ppnK* gene encoding more active PpnK enzymes (NAD+ kinase, catalyzing the conversion of NAD+ to NADP+) was confirmed to be associated with the growth advantages in a fast-growing *Synechococcus* UTEX 2973 strain[31]. Possibly by following similar mechanisms, three genes encoding the NAD(P)+ transhydrogenase were up-regulated in the strain carrying an NC2 mutation, which might also promote the rapid intra-turnover of NAD(P)H pool and ensure the supply of NADP+ to accept photosynthetic electron flow. Taken together, the above mechanisms might benefit the efficiency and flexibility of energy transfer and utilization in the NC2 strain and provide explanations for the growth advantages under NLNT conditions. Furthermore, under HLHT stress, several protective mechanisms were activated in the strain carrying the NC2 mutation, including the reduction of energy absorption (decreased amount of chlorophyll and antenna protein, alteration in photosystem I and photosystem II components), enhancement of energy transfer (ferredoxin, H+ translocating NADPH-NAD+ to NADH-NADP+ transhydrogenase, and components of the oxidative phosphorylation pathway), and increased protein synthesis. Some of the protective or responsive mechanisms could also be observed in WT *Synechococcus* cells under HLHT stress, while the introduction of the NC2 mutation might further enhance these protective effects.

Utilizing the engineered hypermutation system, HLHT tolerance of *Synechococcus* photosynthesis was improved and the relative mechanism was deciphered in this study. Although the improved phenotypes were mainly associated with known or predictable physiological and biochemical mechanisms, genetic evolutionary paths for cyanobacteria cells to adapt to the HLHT stress were identified. More detailed mechanisms associated with genomic mutations and metabolic activities should be further studied, and the functional mutations identified in this study could be promising targets for the optimization of the photosynthetic machinery in the future.

## Methods

### Cyanobacteria strains and culture conditions

*S. elongatus* PCC 7942 and *Synechocystis* sp. PCC 6803 strains used in this study were obtained from Professor Xudong Xu of the Institute of Hydrobiology, Chinese Academy of Sciences. The information on the strains constructed and used in this study is listed in Supplementary Data 12. Cyanobacterial cells were cultivated in BG11 medium supplemented with the corresponding antibiotics (50 μg/mL kanamycin, 25 μg/mL spectinomycin, 5 μg/mL gentamycin, and 10 μg/mL chloramphenicol) as necessary. For routine passages and growth evaluation of the DNA replication fidelity defective mutants, cells were cultivated in flasks shaken under 50 μmol photons/m$^2$/s white fluorescent light at 30 °C. For the hypermutation evolution process and growth evaluation of the evolved strains, cells were inoculated into the liquid culture broth at an OD$_{730}$ of 0.05 and incubated in the multi-cultivator MC1000 (Photon System Instruments, Czech Republic) bubbled with 3% (volume to volume) CO$_2$ (unless stated otherwise); the temperature and light intensity were set as required. For cultivation and evaluation on solid plates, BG11 culture medium was supplemented with 1.5% agar and antibiotics, and the cells were cultivated in tailored incubators and the temperature and light intensity were set as required.

### Construction of plasmids and cyanobacterial strains

Information regarding the plasmids constructed and used in this study is listed in Supplementary Data 2. All the plasmids were constructed using Gibson Assembly by ClonExpress® Ultra One Step Cloning Kit (Vazyme, Nanjing, China) utilizing *E. coli* DH5α (Trans-Gen, Beijing, China). Homologous recombination strategy was routinely adopted for gene knockouts, overexpression, or point-mutation manipulations of the cyanobacterial chromosomes, and the respective plasmids carrying the upstream and downstream homologous fragments as well as the inserted cassettes were assembled into recombinant plasmids and transformed into *S. elongatus* PCC 7942 and *Synechocystis* sp. PCC 6803 cells, following previously described protocols[23] with some modifications. Briefly, 1 mL cells with an OD$_{730}$ of 1.0 to 2.0 were centrifuged at 5000 × *g* for 5 min and then resuspended in 250 μL fresh BG11 medium. Next, 200 ng of each plasmid was added, and the mixture was incubated in the dark at 30 °C overnight. Then the cells were applied to BG11 plates supplemented with the corresponding antibiotics. Plasmids pUC19, pBR322, and pEASY-Blunt were used as backbones for recombinant plasmid construction; to overexpress the genes of interest, the cassettes were integrated on neutral site I (NSI) and neutral site II (NSII) in *Synechococcus* and *slr0168* site in *Synechocystis*. Most of the endogenous fragments were amplified from the genomic DNA of the WT or evolved strains. Primers used for plasmid construction in this study are listed in Supplementary Data 13.

To identify the functional mutations that induce improved HLHT tolerance in *Synechococcus*, fragments containing specific genomic mutations were amplified from the evolved strains and cloned into the pUC19 plasmid. The recombinant strain was transformed into *Synechococcus* using a previously developed survival-selection procedure[23]. Briefly, after being incubated in the dark at 30 °C overnight, the mixture of plasmid and cells during the transformation process were applied to BG11 plates and cultured in an incubator at 500 μmol photons/m$^2$/s (warm white light source) and 44 °C. for 4 days. Then transformants were picked, streaked on BG11 plates, and incubated at 44 °C and 500 μmol photons/m$^2$/s for 3 days. Wild-type *Synechococcus* and pSS4 carrying 0336 mutation (AtpA-C252Y) were used as negative and positive controls, respectively. Transformants that survived after the two-step screening were considered true HLHT-tolerant colonies. Then, the SNP-containing DNA fragments were amplified from the genomic DNA of these transformants for Sanger sequencing (as shown in strategy I of Fig. 5a). Additionally, in strategy II to identify the functional mutation, antibiotic resistance genes were inserted into the nearby intergenic sequence of the specific mutations (details are shown in Supplementary Fig. 11).

### MMS mutagenesis

*Synechococcus* cells in the mid-exponential phase were harvested and resuspended in fresh BG11 with 10$^8$ cells/ml. MMS was added as required and after 1 min incubation, cells were washed with fresh BG11 twice. Then cells were diluted for lethality and harvested for mutation rate determination or cultivated under HLHT selective conditions.

### Lethality determination

WT cells in mid-log phase with 10$^8$ cells/mL were incubated with up to 3 v% MMS for 1 min. Cells were then washed with fresh BG11 twice and 10$^4$ cells (0.05 v% to 2 v%) or 10$^5$ cells (1 v% to 3 v%) were plated on solid BG11. The counted colonies were divided by total plated cell numbers (counted by an automated cell counter, Countstar, Shanghai, China) to calculate the survival rate.

### Evaluation of relative mutation rates of *Synechococcus* genome replication

For pre-cultivation, *Synechococcus* cells were inoculated in BG11 medium at an initial OD$_{730}$ of 0.2–0.3 in column photobioreactors and cultivated under 150 μmol photons/m$^2$/s white fluorescent light at 30 °C, bubbled with air for 3 d. The pre-cultivation broth was re-inoculated into fresh BG11 medium and cultivated under the same conditions until the mid-exponential phase. Especially for MMS-treating cells, before evaluation, they were inoculated in BG11 medium at an initial OD$_{730}$ of 0.2 in a flask under 50 μmol photons/m$^2$/s white fluorescent light at 30 °C or in column photobioreactors and cultivated in MC1000 under 42 °C and 800 μmol photons/m$^2$/s to calculate the non-lethal mutation rates. The cell numbers were calculated using an automated cell counter (Countstar, Shanghai, China), and approximately 1 × 10$^9$–4 × 10$^9$ cells were collected and plated onto a solid BG11 medium containing 15 μg/mL rifampicin. The rifampin-resistant colonies were counted after 2 weeks of cultivation (ImageJ 1.52a software was used to count the rifampicin-tolerant colonies), and the frequencies of the resistant cells in the original culture broth were calculated to evaluate the mutation rates of *Synechococcus* genome replication. The relative mutation rate of the recombinant strains was calculated by comparing it with that of the WT. For the hypermutation evolution process, cultivation was performed in MC1000 with the temperature and light intensity set as required.

### Isolation of evolved *Synechococcus* strains with improved HLHT tolerance

Cells were cultured in MC1000 under 42 °C and 800 μmol photons/m$^2$/s to logarithmic growth phase for hypermutation evolution or treated with 2 v% MMS for MMS mutagenesis as described in the section of "MMS mutagenesis" and "Evaluation of relative mutation rates of *Synechococcus* genome replication." Then, 10$^9$ cells were harvested and inoculated on BG11 plates for screening in an incubator at 500 μmol photons/m$^2$/s (warm white light source) and 44 °C. After 4 days of cultivation, the surviving colonies were inoculated on fresh BG11 plates for re-screening in the incubator. Finally, the tolerant mutants (that survived the re-screening process) were transformed with pHS81 to restore the coding sequence of *mutS* and to remove the additional *recA*.

### Adaptive evolution for HLHT tolerance

*Synechococcus* cells were inoculated in BG11 medium at an initial OD$_{730}$ of 0.2–0.3 in column photobioreactors and cultivated under 150 μmol photons/m$^2$/s white fluorescent light at 30 °C, bubbled with air for 3 days. Then, the pre-cultivation broth was re-inoculated into fresh BG11 medium and cultivated under the same conditions until the mid-exponential phase. WT with 10$^8$ cells/mL was incubated with 2 v% MMS for 1 minute. Cells were then washed with fresh BG11 twice. And cells

(HS84, WT, and WT treated with MMS) were inoculated in BG11 medium at an initial $OD_{730}$ of 0.2 in column photobioreactors and cultivated in MC1000 under 42 °C and 800 µmol photons/m$^2$/s. Then, cells cultured in the mid-log phase were inoculated in BG11 medium at an initial $OD_{730}$ of 0.05 in column photobioreactors and cultivated in MC1000 under 42 °C and 1500 µmol photons/m$^2$/s. After that, cells cultured in the mid-log phase were inoculated in BG11 medium at an initial $OD_{730}$ of 0.05 in column photobioreactors and cultivated in MC1000 under 42 °C and 2000 µmol photons/m$^2$/s and 45 °C and 2000 µmol photons/m$^2$/s.

### Whole-genome re-sequencing

For genome re-sequencing of HS121, HS122, and HS123, genomic DNA was isolated and analyzed for quality and concentration using a Nanodrop ND-1000 system (Thermo Scientific, US) and Qubit Fluorometer (Thermo Scientific, US). Subsequently, the quantified DNA samples were fragmented, blunted, modified with 3'-A overhangs, ligated to Illumina's standard sequencing adapters, and amplified using PCR. The library was sequenced as paired-end reads using an Illumina HiSeq 4000 sequencer at Allwegene (Beijing, China). The reference sequence (*S. elongatus* PCC 7942, FACHB-805) was obtained from GenBank [https://ftp.ncbi.nlm.nih.gov/genomes/all/GCA/000/012/525/GCA_000012525.1_ASM1252v1/] for read mapping using BWA software (v0.7.8), and SAMTOOLS (v0.1.18) was used to sort the results of read mapping and to mark duplicate reads.

For whole-genome re-sequencing of HS137–HS156, genomic DNA was extracted using the SDS method, and the harvested DNA was detected using agarose gel electrophoresis and quantified using Qubit® 2.0 Fluorometer (Thermo Scientific, US). The DNA sample was fragmented, end-polished, A-tailed, and ligated with a full-length adapter for Illumina sequencing with further PCR amplification. Then the PCR products were purified (AMPure XP system) and libraries were analyzed for size distribution by Agilent2100 Bioanalyzer and quantified using real-time PCR. The whole genome was sequenced using the Illumina NovaSeq 6000 at Beijing Novogene Bioinformatics Technology Co., Ltd. The reference sequence (*S. elongatus* PCC 7942, FACHB-805) was obtained from GenBank [https://ftp.ncbi.nlm.nih.gov/genomes/all/GCA/000/012/525/GCA_000012525.1_ASM1252v1/] for read mapping using the BWA software (V0.7.8). SAMTOOLS (v0.1.18) was used to detect individual SNPs and the InDels of small fragments (<50 bp), as well as to analyze the variation in SNP/InDel in the functional regions of the genome. Finally, DNA fragments containing SNPs and InDels were amplified from the genomic DNA and further verified by Sanger sequencing.

### Variant analysis

Hierarchical clustering was used to represent the divergence of mutation sites between two different samples using a previously developed method with minor modifications[30]. Briefly, all the mutations in the 23 evolved strains were denoted on the abscissa, and numbers 1 and 0 were used to denote the presence or absence of a mutation as shown in Supplementary Data 1. Thus, each evolved strain formed a mutated vector. For all the mutations of the two strains, namely, vectors *a* and *b*, the distance of *a* and *b* were defined as the following equation:

$$\text{Distance} = 1/(a \times b + 1) \qquad (1)$$

Code used for hierarchical clustering is available at Github [https://github.com/jibeilindong/Drawtree/blob/main/drawtree.r].

### Room temperature fluorescence kinetics

Approximately $1.7 \times 10^9$ cells in 3 mL were dark-acclimated for 20 min at room temperature (23 °C). And the fluorescence parameters $F_v/F_m$ (maximum quantum yield) of PSII were measured using a Dual-PAM 100 (Heinz Walz) (Dual PAM v1.19 software). A measure light intensity of 19 µE/m$^2$/s was used and the saturating light pulse was 6000 µE/m$^2$/s (200 ms). The maximum chlorophyll fluorescence after adding 20 µM DCMU with AL (54 µE/m$^2$/s) was recorded as the maximum $F_m$.

### Real-time qPCR analysis

*Synechococcus* cells in the mid-exponential phase were sampled for total RNA extraction using a bacterial RNA extraction kit (Vazyme, Nanjing, China). The RNA samples were then reverse transcribed into cDNA using HiScript III RT SuperMix for qPCR (Vazyme, Nanjing, China). Quantitative RT-PCR was performed on a LightCycler 480 Sequence Detector (Roche, Basel, Switzerland) (LightCycler 480 software 1.5) based on SYBR Green I fluorescence (ChamQ Universal SYBR qPCR Master Mix-Vazyme, Nanjing, China) and analyzed using the $2^{-\Delta\Delta Ct}$ method[62]. Primers used for qPCR are listed in Supplementary Data 13.

### Western blot analysis

The western blot assays were carried out in accordance with standard protocols[63]. Briefly, *Synechococcus* cells were centrifuged at 4 °C and resuspended in 50 mM Tris-HCl buffer (pH 8.2). An appropriate amount of quartz sand was added to the sample, and the cells were lysed through vortex oscillation at low temperature. Subsequently, the mixture was centrifuged at $10,000 \times g$ for 20 min to remove quartz sand and debris, and the protein concentration in the supernatant was quantified using the Bradford Protein Assay Kit (Beyotime, Shanghai, China). Then, 30 µg protein was used and separated by SDS-PAGE and visualized by western blot with anti-shikimate kinase antibody. The rabbit polyclonal primary antibody against the shikimate kinase of *Synechococcus* was prepared and purchased from Atagenix Technology Co., Ltd (Wuhan, China). Briefly, the recombinant protein with a GST tag, *Synechococcus* shikimate kinase, and His tag with a linker was expressed in *E. coli* and purified to prepare rabbit-sourced antibodies. The primary antibody was verified by indirect ELISA and western blot. Thereafter, goat-anti-rabbit IgG (HRP conjugated antibody, Solarbio, Beijing, China) (1:2,000 dilution) was applied to hybridize with the primary antibody (1:1000 dilution). The AEC Peroxidase Substrate Kit (Solarbio, Beijing, China) was used for color rendering.

### RNA-seq analysis

RNA sequencing was performed by Novogene Bioinformatics Technology Co., Ltd. (Beijing, China). Total RNA was isolated using a Tiangen RNA extraction kit (Tiangen, Beijing, China), according to the manufacturer's instructions. RNA integrity was assessed using an RNA Nano 6000 assay kit (Bioanalyzer 2100, California, US). The mRNA was purified from the total RNA, using probes to remove rRNA, to construct sequencing libraries. Library quality was assessed using the Agilent Bioanalyzer 2100 system. Clustering of index-coded samples was performed on a cBot Cluster Generation System using the TruSeq PE Cluster Kit v3-cBot-HS (Illumina, California, US). After cluster generation, the library preparations were sequenced on the Illumina NovaSeq 6000 platform and 150 bp paired-end reads were generated. Raw data were processed using in-house Perl scripts to generate high-quality clean data. The reference genome and gene model annotation files (*S. elongatus* PCC 7942 and FACHB-805) were obtained from GenBank [https://ftp.ncbi.nlm.nih.gov/genomes/all/GCA/000/012/525/GCA_000012525.1_ASM1252v1/] for read mapping using Bowtie2 (v2.3.4.3). HTSeq (v0.9.1) was used to count the read numbers mapped to each gene, and then the FPKM of each gene was calculated. Differential expression analysis of the two groups (three biological replicates per condition) was performed using the DESeq R package (v1.20). The resulting $p$-values were adjusted using the Benjamini and Hochberg's approach to control the false discovery rate. Genes with an adjusted $p$ value <0.05 found by DESeq were assigned as differentially expressed.

## Parallel reaction monitoring (PRM) analysis

To verify the protein expression levels obtained by label-free analysis, the expression levels of selected proteins were determined by LC-PRMMS analysis at Shanghai Applied Protein Technology Co., Ltd. Further quantification was performed using the label-free protocol and peptide retention time calibration mix (Thermo Scientific, US) stable isotopic peptides were added to each sample as an internal standard reference. Tryptic peptides were loaded on C18 stage tips for desalting prior to reversed-phase chromatography on an Easy nLC-1200 system (Thermo Scientific, US). One-hour liquid chromatography gradients with acetonitrile ranging from 5 to 35% in 45 min were used. PRM analysis was performed on a Q Exactive HF mass spectrometer (Thermo Scientific, US). Methods optimized for collision energy, charge state, and retention times for the most significantly regulated peptides were generated experimentally using unique peptides of high intensity and confidence for each target protein (Supplementary Data 4). The mass spectrometer was operated in positive ion mode and with the following parameters: The full MS1 scan was acquired with the resolution of 70,000 (at 200 $m/z$), automatic gain control (ACG) target values $3.0 \times 10^{-6}$, and a 200 ms maximum ion injection times. Full MS scans were followed by 20 PRM scans at 35,000 resolution (at $m/z$ 200) with AGC $3.0 \times 10^{-6}$ and a maximum injection time 200 ms. The targeted peptides were isolated with a 1.5 Th window. Ion activation/dissociation was performed at a normalized collision energy of 27 in a higher energy dissociation (HCD) collision cell. The raw data were analyzed using Skyline (MacCoss Lab, University of Washington)[64] where signal intensities for individual peptide sequences for each of the significantly altered proteins were quantified relative to each sample and normalized to a standard reference. Proteins with a $p$ value <0.05 found by two-tailed unpaired Student's $t$ test were assigned as differentially expressed. The $p$ values were listed in Supplementary Data 4.

## P700 analysis

For evaluation of CEF activity, P700$^+$ re-reduction and the redox kinetics of P700 were determined according to previously described methods[65,66] with some modifications. Briefly, *Synechococcus* cells were cultured in BG11 medium at an initial OD$_{730}$ of 0.05 in column photobioreactors and cultivated in MC1000 under 30 °C and 50 μmol photons/m$^2$/s bubbled with air. Cells in the mid-log phase were harvested and adjusted to approximately $6 \times 10^9$ cells in 3 mL and were dark-acclimated for 20 min at room temperature (23 °C). P700 redox was monitored after the termination of AL illumination (2800 μE/m$^2$/s for 35 s) under a background of FR light (Intensity level 20) using Dual-PAM 100 (Heinz Walz) (Dual PAM v1.19 software). The P700 levels were normalized by equating the absorbance minimum after the termination of AL illumination to 0 and equating the absorbance maximum of 3 s before the termination of AL illumination to one. The re-reduction of P700$^+$ in darkness was measured by monitoring absorbance changes at 830 nm and using 875 nm as a reference. After dark-acclimated for 20 min, 20 μM DCMU was added to the cultures before the measurement. The P700 was oxidized by FR light (Intensity level 20) for 40 s, and the subsequent re-reduction of P700$^+$ in the dark was monitored. The curves were normalized by equating the absorbance maximum of 1 s before termination of FR light to one. The initial rate of P700$^+$ re-reduction was acquired from normalized curves by calculating the slope of the initial values after FR was turned off and the R$^2$ was more than 0.99.

## Determination of photosynthetic O$_2$ evolution and dark respiration

*Synechococcus* cells in the mid-exponential phase were harvested and resuspended in fresh BG11 at an optical density at 730 nm of 1.0.

The respiration rate was measured under dark for 2 min at 30 °C or 41 °C and the net photosynthetic O$_2$ evolution rate at each light level (RGB light source) was measured with the YZQ-201A photosynthetic instrument (YZQ-201A13 software) in the presence of 10 mM NaHCO$_3$. And the total photosynthetic O$_2$ evolution rate was calculated as the net photosynthetic rate plus the respiration rate. For the measurement of PSII-mediated O$_2$ evolution, 1 mM 1,4-benzoquinone and 1 mM K$_3$Fe(CN)$_6$ were added. To monitor the repair of PSII, strains in the mid-log phase cultured under NLNT (30 °C and 500 μmol photons/m$^2$/s) conditions were exposed to HLHT conditions (42 °C and 2500 μmol photons/m$^2$/s) with or without lincomycin (200 μg mL$^{-1}$) and the PSII-mediated O$_2$ evolution rates were detected every 30 min.

## Determination of ROS in *Synechococcus* cells

The ROS levels in *Synechococcus* cells were measured using the fluorescence probe DCFH-DA and a Reactive Oxygen Species Assay Kit (Beyotime, Shanghai, China). Approximately $2 \times 10^8$ *Synechococcus* cells were collected and resuspended in 1 mL of fresh BG11 medium and then supplemented with DCFH-DA to a final concentration of 10 μmol/L. After incubation at 37 °C in a shaker for 20 min, the cells were washed three times with fresh BG11 medium to fully remove the unabsorbed DCFH-DA. Finally, 200 μL samples were added to 96 well plates, and the fluorescence intensities (emission wavelength of 525 nm) were detected at an excitation wavelength of 488 nm using Tecan Infinite M200 PRO (Tecan, Austria).

## Determination of chlorophyll contents of *Synechococcus* cells

*Synechococcus* cells cultivated under NLNT and HLHT conditions were collected after 48 h of cultivation, and the chlorophyll $a$ content was determined using the methanol extraction method[67] with some modifications. Two mL of cultures were collected and resuspended in 2 mL of methanol in the dark at −20 °C overnight. After centrifugation at $10,000 \times g$ and 4 °C for 5 min, the absorbance of the supernatant was measured at 665 and 720 nm using Tecan Infinite M200 PRO (Tecan, Austria). Chlorophyll $a$ content was calculated using the following equation:

$$\text{Total chlorophyll } a \text{ content } (\mu g/mL) = 12.9447 \times (A_{665} - A_{720}) \quad (2)$$

## Determination of protein content in *Synechococcus* cells

Protein content in *Synechococcus* cells was determined using a previously described Kjeldahl nitrogen determination method[68] with minor modifications. Briefly, 0.2 g of lyophilized sample powder, 1.0 g CuSO$_4$, 3.0 g K$_2$SO$_4$, and 10.0 mL H$_2$SO$_4$ were added to a digestion tube. The mixture was heated in the following order: at 200 °C for 20 min, 300 °C for 20 min, and 420 °C for 90 min in graphite digestion apparatus (Hanon, Jinan, China). After cooling to 25–30 °C, the mixture was analyzed using an automatic Kjeldahl nitrogen analyzer (Hanon, Jinan, China). Finally, the protein content was calculated from the total N content by multiplying it by a conversion factor of 6.25.

## Determination of glycogen content in *Synechococcus* cells

Intracellular glycogen content was calculated as previously described[69] with minor modifications. Briefly, *Synechococcus* cells were collected after centrifugation at $10,000 \times g$ for 15 min. The pellet was washed three times with ddH$_2$O, resuspended in 30% (w/v) KOH, heated at 95 °C for 2 h, and finally, ice-cold ethanol was added to a final concentration of 70–75% (v/v). The mixture was cooled overnight at −20 °C, then centrifuged at $10,000 \times g$ for 15 min to collect the glycogen pellet. Glycogen pellets were washed twice consecutively with 70% ethanol (v/v) and 98% ethanol (v/v) and then dried by vacuum centrifugation. The resulting glycogen was suspended in 100 mM sodium

acetate and digested with amyloglucosidase (Novozymes). To measure the amount of glucose produced in the glycogen solution, a sucrose/D-glucose assay kit (Megazyme, Ireland) was used.

**Determination of Y, F, and W frequency of *Synechococcus* cells**
DeepTMHMM was used for the prediction of the topology of both alpha-helical and beta-barrel transmembrane proteins[70] and the 2657 proteins (UniProt ID: UP000002717) in *Synechococcus* cells were predicted. The frequency of Y, F, and W in each protein was counted using R programming language. Code used for analyzing the frequency of tyrosine, phenylalanine, and tryptophan in proteins is available at Github [https://github.com/jibeilindong/Drawtree/blob/main/AA_frequency.R].

**Reporting summary**
Further information on research design is available in the Nature Portfolio Reporting Summary linked to this article.

## Data availability
Raw data for whole-genome re-sequencing are available at NCBI under accession number PRJNA846529. Raw data for RNA-seq analysis are available at NCBI under accession number PRJNA847037. The reference genome and gene model annotation files (*S. elongatus* PCC 7942 and FACHB-805) for whole-genome re-sequencing and RNA-seq analyses were obtained from GenBank [https://ftp.ncbi.nlm.nih.gov/genomes/all/GCA/000/012/525/GCA_000012525.1_ASM1252v1/]. Source data are provided with this paper.

## Code availability
Code used for hierarchical clustering in the variant analysis is available at Github [https://github.com/jibeilindong/Drawtree/blob/main/drawtree.r]. Code used for analyzing the frequency of tyrosine, phenylalanine, and tryptophan in proteins is also available at Github [https://github.com/jibeilindong/Drawtree/blob/main/AA_frequency.R].

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

## Acknowledgements

The research was supported by the National Key Research and Development Program of China (Grant number 2021YFA0909700, to X.L. and G.L.), the Youth Innovation Promotion Association CAS (to G.L.), the National Natural Science Foundation of China (Grant number 32070084 to G.L., 32270103 to G.L., 32271484 to X.L.), the DNL Cooperation Fund, CAS (DNL202014, to G.L.), and the Shandong Taishan Scholarship (to X.L. and G.L.).

## Author contributions

G.L. and X.L. designed the research; H.S., G.L., Y.M., W.L., R.C., S.Z., J.S., and D.F. performed the research; H.S., G.L., and X.L. analyzed the data; and H.S., G.L., and X.L. wrote the manuscript.

## Competing interests

The authors declare no competing interests.
