## [Peer Review File · Nature Communications]

Engineered hypermutation adapts cyanobacterial photosynthesis to combined high light and high temperature stressReviewers' Comments:

Reviewer #1:

Remarks to the Author:

General:

The authors present a study in which they use an "endogenous" mutagenesis system to identify mutations that confer enhanced tolerance to simultaneous high light and heat. The mutagenesis system is interesting. The fact, that the authors study a combined stress instead of the single components separately and then comparing it with the combined stress is clearly a flaw. Moreover, the authors perform a 4 day selection which is more a classical forward genetics approach like published repeatedly before for high light tolerance in *Chlamydomonas* rather than adaptation which would involve then the accumulation of several mutations. The authors also oversell their approach by claiming that ALE takes years. This is not true since several months usually suffice – even if the first study in *Synechocystis* that the authors refer to took longer because they wanted probably to be on the safe side having no prior knowledge whether the approach is feasible or not. Currently, there are also cyanobacterial (and some green algal) strains available such that ALE can be done in few weeks with or without additional mutagenesis. Moreover, the approach described here also benefited apparently from the enhanced mutagenesis rate induced by the combined stress. The same result might have been obtained then by gradually increasing light and temperature in a couple of weeks, maybe even without mutagens, and this could be also called "efficient". Also, the approach of the authors involves repair of the hyper mutator which markedly decrease the "efficiency" of their approach. The last part of the results is rather descriptive such that one could make two publications out of this. One introducing the technological advance (which should be done in comparison with classical ALE) and a follow-up one describing one or two interesting mutations.

Therefore, to fully support their claims the authors need to:

1. Study single stressor (HL, HT) and compare it with the combined stress (HL+HT)
2. Truly and objectively assess the advantage of endogenous mutagenesis by appropriate controls. Like making one round of EMS or UV mutagenesis and compare it with their system.
3. Cultivation conditions are dubious. 500 mikro mol is not normal light intensities but already stressful. For this reason, authors probably used 150 mikromol for pre-cultures (line 624)?
4. Carefully check the text for ambiguous phrasing. There are many, and a part of them is listed below.

Details:

2: Title. "acclimates". Conventionally, acclimation refers to physiological adjustments and adaptation to genetic adjustments. Title is misleading.

35. The authors mean simultaneous application of high light and high temperature. They should write it this way

36: Previous attempts might have been time-consuming but they were certainly not inefficient.

36: "phototrophs" wrong term here.

38: cyanobacterial strain

42: again "acclimation" when "adaptation" would be more meaningful

44: HIGH temperatures and light intensities

45-47: Correct, but this was not the first study to show this.

52: "most active"? in which sense?

55: "artificial domestication". The entire sentence is not really meaningful and the term "artificial domestication" makes little sense. Moreover, photosynthesis came for free during domestication.

58-60: needs rephrasing

61: "high light and temperatures are major stresses that occur simultaneously". This is not 100% right. Actually, plants are used to the simultaneous presence of LH and HT and shut down photosynthesis during these periods, also to avoid water loss. The problem is the combination of cold and high light, like happening in the morning of spring and autumn. This is really harmful since light reactions run hot and the dark reactions are still slow.

63: functional range of photosynthesis: what does this mean?

67/68: See above. I disagree. The study would be much more useful, if the two single components (HL or HT) and the double stress (HL + HT) had been analysed. Only analysing the combined stress is much less informative. Moreover, having studied HL separately would have allowed comparison with the previous study that the authors mention in *Synechocystis*.

70: "putative" can be deleted.

71-73. also algae have short life cycles and rapid growth.

82: authors omitted to cite literature on HT ALE: Tillich et al. 2012 and 2014 for HT ALE.

83: ideal mutants. No, this is not ideal or if this is ideal then many more combinations are equally ideal.

88: 10 to the three magnitude cell replication events. Rephrase. Unclear.

188: 4 days corresponds to how many generations? Is this adaptive evolution or just classical forward genetics in which single mutations are identified rather than epistatic series of mutations?

194: the authors selected three colonies and later many more (239-240). That might reflect the original history of the experiments but there is not need to write it this way because it is confusing at the end. This sound more like a lab book in this part of the manuscript.

197: 30 C and 500 mikromol light cannot be considered as "normal conditions".

243: heterozygous? Diploid organisms? What do you mean. Mention copy number of DNA.

244: what are repeated mutations?

281: here and many times again. Acclimation is not adaptation and vice versa.

335. scarifying. Rephrase.

412: energy-coupled electron transfer between NADH and NADPH: explain.

472: this is not correct. Any ALE can be done in a couple of months, in particular if using fast-growing UTEX strains like the ones described recently by the Pakrasi lab.

492: Previous study with HL tolerance in *Synechocystis* showed already that single mutations can do (part of) the job. So, this is not new.

536-541: is the overrepresentation of aromatic amino acids maybe due to the fact that they are hydrophobic and that photosynthetic proteins are mostly membrane proteins? Sometimes explanations are more obvious than thought. Make a control by analysing separately soluble and membrane proteins.

574-576: disagree. The approach might be faster (maybe not if hyper mutator has to be removed individually in each clone) but the approach is not new. At the end, the mutations do not differ from the ones found by ALE.

987: normally you need DCMU to determine Fm. Could not find this in Methods.

Reviewer #2:

Remarks to the Author:

NatComm

Title: Engineered hypermutation acclimates cyanobacteria photosynthesis to high light and high temperature stress

The manuscript of xx reports on improved mutation rates to capture cyanobacteria strains (*Synechococcus*) by identification and knockout of multiple genes involved in DNA replication fidelity. Moreover, the authors show that mutation rates can be further enhanced by exposing cultures to high light and/or high temperature, producing a large number of HLT strains. Many of the "HLT" mutants exhibited mutations causing enhanced expression of shikimate kinase, an enzyme of a major pathway for secondary metabolite production. Furthermore, the researchers overexpressed shikimate kinase in a *Synechocystis* strain, and showed that the overexpression strain also gained tolerance to high light. They also suggest that upregulation of the shikimate pathway may represent a universal strategy for improving tolerance to light and/or temperature stress in cyanobacteria. Last, RNAseq and qPCR were performed to estimate the overall impact on global metabolism, photosynthesis and other regulatory processes. This work is a significant step forward in significantly improving the mutational rate in cyanobacteria and this tool should be of great interest to the scientific community in rapidly engineering novel strains with improved tolerance to a range of climate-relevant stressors. However, I do think that the discussion relied heavily on the RNAseq results to speculate on a lot of physiological changes in the mutants. It would be nice to see some additional measurements to support the changes within the photosynthetic apparatus.

Line 35 – I would suggest adding "combined" to emphasize that you are referring to the combined HL/HT stress (same comment throughout the manuscript). I also suggest that the acronym "HLT" is a little confusing since many publications use "LT" for low temperature – you could consider using "HLHT"

Line 70 – why do you refer to cyanobacteria as "putative" ancestors of chloroplasts?

Introduction – it would be nice to see a good rationale for why it specifically important to isolate high light/high temperature – tolerant mutants

Line 84 – I don't think it's accurate to say that photosynthetic tolerance to HL and HT stress is poorly understood.

Line 399 – the RNA seq results suggesting remodelling of key photosynthetic electron transport chain components was very interesting, particularly those associated with energy homeostasis. Did the authors consider supporting these findings with some photosynthetic measurements, such as oxygen evolution, chlorophyll fluorescence, or P700 spectroscopy? I think some basic photosynthetic measurements (beyond chlorophyll estimations) would be a significant contribution to the impact of this manuscript.

Line 520 – The finding that a major enzyme of the shikimate pathway can confer significant increases in stress tolerance is very interesting. I would like to see more discussion – this paragraph is pretty superficial regarding the potential impacts on cellular physiology and stress tolerance.

Reviewer #3:

Remarks to the Author:

This manuscript describes the use of a creative approach to identify mutations in the cyanobacterium *Synechococcus* sp. PCC 7942 that confer high light and temperature (HLT) stress tolerance. The authors use a strategy of perturbing the fidelity of DNA replication combined with environmental stress that increases ROS production, which allowed direct and rapid selection of mutants that survive lethal HLT stress in a process that is much faster than conventional adaptive laboratory evolution. They nicely showed that two types of mutations on their own could confer HLT tolerance, whereas others can improve HLT tolerance in combination. One of the two major mutations (AtpA-C252Y) had been found previously, but the other (NC2) affected expression of shikimate kinase. They convincingly showed that overexpression of shikimate kinase confers HLT tolerance, even in a different species of cyanobacterium, and that this has a large and pleiotropic effect on the transcriptome of *Synechococcus*. Overall, I found the approach and results in this manuscript to be interesting and convincing.

Major comment:

1. The NC2 mutation was identified as a major route to HLT tolerance, however the molecular mechanism of how the upregulation of shikimate kinase via NC2 mutation confers HLT tolerance was not determined. The authors' speculation (lines 539-541) "that an enhanced supply of aromatic amino acids might promote the synthesis of essential photosystem components" is intriguing and supported by finding other mutations related to protein synthesis (i.e., mutations affecting GTP synthesis and EF-Tu) that further enhance the HLT tolerance of NC2 (Fig. 5). PSII repair via D1 protein turnover and replacement is known to be very important for photoprotection in cyanobacteria, so it was surprising that the authors did not test this experimentally. They could examine rates of PSII photoinhibition and recovery in the presence or absence of lincomycin, rates of D1 synthesis, and levels of D1 protein.

Minor comments:

2. Title: "adapts" would be preferable to "acclimates", because this is a heritable change. This should also be changed in the main text.

3. line 52: Clarify what is meant by "most active".

4. line 70: Cyanobacteria are not "a putative ancestor of chloroplasts". Extant cyanobacteria share a common ancestor with chloroplasts.

5. The manuscript would benefit from copyediting to correct some instances of English usage. Here are just two examples, but there are others:

line 312: change "Complying" to "Consistent"

line 335: What is meant by "scarifying"?

Point-to-point replies to the reviews of manuscript NCOMMS-22-28713-T

REVIEWER COMMENTS

Reviewer #1 (Remarks to the Author)

General:

The authors present a study in which they use an "endogenous" mutagenesis system to identify mutations that confer enhanced tolerance to simultaneous high light and heat. The mutagenesis system is interesting. The fact, that the authors study a combined stress instead of the single components separately and then comparing it with the combined stress is clearly a flaw.

RESPONSE: Thanks for the comments. We agreed that the comparisons between evolved mutants and mechanisms tolerant to single stresses and the combined stress (e.g., HLHT versus HL or HT) would be very interesting and could provide much meaningful information about microbial & plant physiology, metabolism, and evolution. However, we believed that such experiments and analysis could form another interesting story and would be beyond the aim and scope of this work.

High light and high temperature are both severe environmental stresses that impair photosynthetic activities of plants and algae, and the combined effects of the two stresses (HLHT) are also posing serious threats to agriculture and other photosynthesis derived economy activities. Lots of research have been performed to explore specific responsive or protective mechanisms toward a single stress (high light or high temperature) in photoautotrophs, including several works utilizing cyanobacteria as models. However, relative understandings about the combined HLHT of photoautotrophs are quite limited. Regarding the strain improvement efforts with cyanobacteria or eukaryotic microalgae, mutants or recombinant strains with improved tolerances to both HL and HT are rarely reported, which also restricted the explorations on tolerance mechanisms of the combined HLHT stress. The focus of this manuscript is to develop an effective evolutionary engineering approach and a promising paradigm to promote the address of the challenge of improving complex phenotypes in cyanobacteria. The inherent logic is "development of a novel method -> improvement of a phenotype -> identification of contributing SNPs -> clarification of the physiological mechanisms". We selected HLHT tolerance of photosynthesis as the target phenotype to improve with cyanobacteria and to check the effectiveness of the novel approach (engineered hypermutation system) developed in this work (we have added the word "combined" in front of the "high light and high temperature" in the revised title).

Following the reviewer's suggestion, we have designed and initiated experiments for parallel evolution of tolerant phenotypes toward single stresses (HL and HT) and combined stress (HLHT) with the newly developed engineered hypermutation system as well as other classical random mutagenesis approaches and adaptive laboratory evolution strategy. The experiments are currently underway and we hope that more systematic presentations as a new and separate story could be given in future.

Moreover, the authors perform a 4-day selection which is more a classical forward genetics approach like published repeatedly before for high light tolerance in *Chlamydomonas* rather than adaptation which would involve then the accumulation of several mutations.

RESPONSE: Thanks for the comments. As noted by the reviewer and mentioned in the abstract and main text of our manuscript, we have essentially adopted a forward genetics research paradigm for improving HLHT tolerance of *Synechococcus* and exploring the related mechanisms; meanwhile, we also adopted the reverse genetics strategy to validate the functions of the SNPs mined in the evolved mutants. Forward genetics approaches have been widely utilized in improving and deciphering cellular robustness and fitness phenotypes, due to the complex mechanisms and limited understandings of such traits. The classical chemical/physical random mutagenesis approaches and the long-term adaptive laboratory evolution approaches, as well as the combination of more than one approach, can all be recognized as the tools supporting forward genetics research, during which the genome mutations would be generated/introduced/accumulated in the offspring populations to expand genetic/phenotypic diversities for screening/selection of mutants tolerant to the stressful conditions ^{1, 2}. For example, Lisa Schierenbeck et al. (the mentioned work; BMC Genomics, 2015³) treated the *Chlamydomonas* with UV exposure to induce genomic mutations for high light tolerant mutants screening, Yoshikawa et al. (Communications biology, 2016⁴) performed long-term/repeated cultivation of *Synechocystis* to accumulate and isolate spontaneous genomic mutations responsible for high light adaptation, and Dann et al. (Nature Plants, 2021⁵) combined adaptive laboratory evolution and the chemical/UV random mutagenesis to improve high light tolerance of cyanobacteria. For the forward genetics research, one of the key steps holding control for obtaining desired mutants and traits, is the generation of genetic & phenotypic diversities, meaning the generation of genomic mutations. In the classical random mutagenesis experiments, a large number of genomic mutations would be artificially induced by chemical mutagens or physical stress (e.g., UV and NTG/MMS/EMS) and positive mutants would be screened out in one go (usually on

solid plates); while in classical adaptive laboratory evolution, the slowly generated spontaneous mutation would be continuously selected and accumulated by gradually enhanced selective pressures (meaning repeated passages in liquid or solid culture medium). An essential aim of this work is to develop a novel genome mutagenesis approach/strategy, that could conveniently and autonomously introduce mutations into cyanobacteria genomes, and such a system could be expected to not only reduce the dependence on traditional mutagenesis manipulations (chemical mutagens and UV, which are generally considered to be biotoxic and carcinogenic) but also to improve the efficiency of adaptive laboratory evolution (due to the significantly improved rates of spontaneous mutations). In the revised manuscript, we have provided data supporting the effects of the engineered hypermutation system on both classical direct screening process and the liquid-passage supported adaptive laboratory evolution process (Supplementary Fig. 9).

The authors also oversell their approach by claiming that ALE takes years. This is not true since several months usually suffice – even if the first study in *Synechocystis* that the authors refer to took longer because they wanted probably to be on the safe side having no prior knowledge whether the approach is feasible or not. Currently, there are also cyanobacterial (and some green algal) strains available such that ALE can be done in few weeks with or without additional mutagenesis.

RESPONSE: Thanks for the comments. We fully accept the reviewer's critical comments regarding our inappropriate presentation of the efficiency of the classical ALE approaches (e.g., in Line 95, the sentences have been rephrased as “The conventional adaptive laboratory evolution (ALE) approach, even enhanced with physical and chemical mutagenesis treatments, usually requires weeks or months to obtain cyanobacteria mutants with improved tolerances to high light or high temperature stress”), and we have revised relative descriptions and presentations in the revised manuscript. Also, as mentioned in the previous response comments, the focus of this work on methodology is to develop a tool that can facilitate and automate the process of continuously introducing random genome mutations in cyanobacteria, which could be expected to increase genome mutation rates and accelerate evolution. Our approach can be conveniently combined with existing methods and techniques, either classical direct screening process on solid plates or ALE strategies through repeated passages in liquid medium, to improve the efficiency and effectiveness of cyanobacterial strain improvement.

Meanwhile, we agreed that the emerging fast-growing microalgae strains could be used for rapid and efficient ALE process. While it is also noteworthy that, the utilization

of “mutagenesis” approaches, whether the classical chemical/physical mutagens or the in vivo mutagenesis strategies, could be expected to further accelerate the evolution process by rapidly increasing the genetic diversities of the offspring. Therefore, enriching the evolutionary engineering toolbox should be as important and valuable for this field as the novel strains.

Moreover, the approach described here also benefited apparently from the enhanced mutagenesis rate induced by the combined stress. The same result might have been obtained then by gradually increasing light and temperature in a couple of weeks, maybe even without mutagens, and this could be also called “efficient”. Also, the approach of the authors involves repair of the hyper mutator which markedly decrease the “efficiency” of their approach.

RESPONSE: Thanks for the comments. As the reviewer pointed out, the engineered hypermutation system developed in this work benefited from the enhancement of mutagenesis strengths induced by the combined stress (at a non-lethal level), and this is one of the main findings and innovations of this work. To develop hypermutable system in cyanobacteria, we systematically evaluated the contributions of potential “mutators” and “anti-mutators” in *Synechococcus* on determining replication mutation rates; and we increased the mutation rates of *Synechococcus* cells by 100-fold through a combinatory manipulation with two genes (*mutS* & *recA*). More importantly, we found for the first time that environmental stress conditions can significantly elevate the mutagenesis strengths of such a system to 1000-fold higher than the control, showing a significant synergetic effect of the replication fidelity deficiency and the environmental stress. In addition, as shown in Fig. 2, the non-lethal environmental stress conditions could also increase replication mutation rates of the WT cells by 3-fold, which would also contribute to our understanding of the role of selective pressures in the classical ALE processes.

We do not doubt the effectiveness and achievements of existing ALE methods as well as classical random mutagenesis methods in improving the complex phenotypes of cyanobacteria as well as other microorganisms, and we also agreed that the HLHT tolerance of *Synechococcus* cells can be enhanced with or without the use of mutagens by gradually increasing the light intensity and temperature of the cultivation, as suggested by the reviewer. While, in this work, we would like to focus on the fact that the hypermutation system we developed could significantly increase the replication mutation rates of *Synechococcus* cells and reduce the dependence on chemical or physical mutagens and related manipulations. As shown in Supplementary Fig. 9, we have compared the mutagenic strengths and evolution effects of the

hypermutation system with a chemical mutagen and the WT control, and results confirmed that at least for the conditions and processes used in this work, the newly developed system possessed advantages in both the solid plates screening and liquid passaging modes.

In addition, as the reviewer pointed out, we performed an additional round of genetic manipulation (simultaneously restoring the *mutS* and removing the additional *recA* copy) to obtain the finally genetically stable mutants, which would cost 4-7 days. Currently, the limitations on efficient and smart synthetic biology approaches (compared with these of *E. coli* or yeast) restricted the development of more “smart” regulation circuits on the replication fidelity machinery in cyanobacteria, and we believed that could be solved in future to develop the next generation of controllable hypermutation system in cyanobacteria. In the meantime, we would like to point out that whether to perform the additional genetic manipulation would not affect the identifications of determining mutations/SNPs in the evolved strains. As shown in Fig, 20 out of the 23 HLHT tolerant strains selected for genome re-sequencing were not manipulated to restore the replication fidelity deficiency, and that did not influence the identification of the functional SNPs in subsequent experiments.

The last part of the results is rather descriptive such that one could make two publications out of this. One introducing the technological advance (which should be done in comparison with classical ALE) and a follow-up one describing one or two interesting mutations.

RESPONSE: Thanks for the comments. As mentioned in previous replies, we appreciated the reviewer’s suggestion, and have designed and initiated experiments for parallel evolution of tolerant phenotypes toward single stresses (HL and HT) and combined stress (HLHT) with the newly developed engineered hypermutation system as well as other classical random mutagenesis approaches. The experiments are currently underway and we hope that more systematic presentations could be given in future. Besides, we have supplemented the comparisons of the engineered hypermutation system with a chemical mutagen (MMS) on mutagenesis strengths and phenotypes improvement in both the solid plate screening process and liquid cultivation process (as shown in Supplementary Fig. 9).

We recognized the flaws of the last part in our previous version of manuscript that the results relied on comparative transcriptomic analysis and appeared too descriptive. We have supplemented additional data about the photophysiology analysis as well as targeted proteome assays to support more convincing explanations of the NC2 effects on *Synechococcus* cells. We preferred to keep the part in the manuscript, as it would

be useful to demonstrate the whole logic and paradigm in which the newly developed approach facilitates improvement and understanding of cellular phenotypes in cyanobacteria.

Therefore, to fully support their claims the authors need to:

1. Study single stressor (HL, HT) and compare it with the combined stress (HL+HT)

RESPONSE: Thanks for the suggestions. As mentioned in the previous response comments, we agreed that the comparisons between the single stress tolerant mechanism and the combined stress tolerant mechanism would be interesting and important, while we thought that would be beyond the aim and scope of this manuscript. The mentioned comparisons could be performed as an individual project and resulted in a systematic and comprehensive presentation. In this work, we chose a combined stress (HLHT) tolerance as the phenotype to improve with the developed hypermutation system because research and understandings about such phenotype are relatively rare comparing with that with single stress tolerance. With the goal of obtaining HLHT tolerant strain and understanding the mechanisms involved, the methodology and paradigm to address the issues would be the focus of this work.

Besides, regarding the single stressor study, we have already performed growth assays of the evolved mutant (HS121, HS122, HS123) and recombinant mutant (HS199, carrying NC2 mutation) under conditions with separately elevated temperature and separately enhanced light intensities (Fig. 3 b-g; Fig. 6 a-b). And the results revealed that the evolved strains tolerant to combined stress (HLHT) also possessed elevated tolerance to a single stressor (HL and HT separately).

2. Truly and objectively assess the advantage of endogenous mutagenesis by appropriate controls. Like making one round of EMS or UV mutagenesis and compare it with their system.

RESPONSE: Thanks for the comments. We appreciated the reviewer's suggestion and performed additional experiments comparing the hypermutation system with random mutagenesis with a chemical mutagen, MMS, which has been utilized in cyanobacteria adaptive evolution (promoting the evolution for thermotolerance). As shown in Supplementary Fig. 5 as well as Fig. R1, WT *Synechococcus* cells were firstly treated with different concentrations of MMS to find out the optimal mutagenesis dose⁶, and the 90% lethal rate (which was usually recognized as suitable for microbial random mutagenesis) was observed in samples treated with 2%.

Fig. R1 Survival rates of *Synechococcus* cells treated with different doses of MMS. WT cells with densities of 10^8 cells/mL were incubated with up to 3% MMS for 1 minute. Cells were then washed with fresh BG11 twice and 10^4 cells were plated on solid BG11. The counted colonies were divided by total plated cell numbers to calculate the survival rate.

The mutation rates were then measured with the 2% MMS treated cells after 7 days of recovery cultivation in fresh BG11 culture broth ⁶. As shown in Fig. R2, replication mutation rates of *Synechococcus* cells were increased by nearly 30-folds (28.5-folds when cultivated with white fluorescent light at 30°C and 50 $\mu\text{mol photons/m}^2/\text{s}$ and 26-folds when cultivated in MC1000 at 42°C and 800 $\mu\text{mol photons/m}^2/\text{s}$). It is noteworthy that although the MMS treatment could improve replication mutation rates and could be expected to accelerate the strain improvement process, it would also cause severe physiological impairments to the *Synechococcus* cells. Without 7 days recovery cultivation in fresh BG11, no colonies could survive and grow on BG11 solid plates containing rifampin; and that would also reduce the process efficiencies when the chemical mutagens were used in adaptive evolution.

Fig. R2 The evaluation of the mutation rates of *Synechococcus* cells treated with 2% MMS. a. Diagram for the evaluation process. WT cells with a density of 10^8 cells/mL were incubated with 2% MMS for 1 minute. MMS-treating cells were then washed with fresh BG11 twice and then inoculated in BG11 medium at an initial OD_{730} of 0.2 in a flask under 50 $\mu\text{mol photons/m}^2/\text{s}$ white fluorescent light at 30°C or in column photobioreactors and cultivated in MC1000 under 42°C and 800 $\mu\text{mol photons/m}^2/\text{s}$ (as used to elevate mutation rates of HS84). Approximately 1×10^9 to 4×10^9 cells were collected and plated onto a solid BG11 medium containing 15

$\mu\text{g/mL}$ rifampin. The rifampin-resistant colonies were counted after two weeks of cultivation to calculate the mutation rate (WT-MMS 30/50 or WT-MMS 42/800). Mutation rates of non-treated WT cells and HS84 cells (as described in the manuscript) would be calculated as controls. b. The relative mutation rates of HS84, WT-MMS 30/50, and WT-MMS 42/800.

We further compared the effects of our newly developed approach and the MMS-induced random mutagenesis method on improving HLHT tolerance of *Synechococcus*. As shown in Fig. R3a&b, when spreading on BG11 agar plates (after recovery cultivation) and cultivated under lethal conditions (44°C and $500 \mu\text{mol photons/m}^2/\text{s}$, warm white light) to the *Synechococcus* cells, no colonies could be obtained from the MMS treating sample, while lots of HS84 colonies survived in the same process and environment. Besides, we also compared the effects of the two methods with liquid cultivation (usually adopted in ALE). As shown in Fig.R3c&d, the *Synechococcus* cells treated with 2% MMS show improved adaption to strong light intensities and high temperature (42°C and $1500 \mu\text{mol photons/m}^2/\text{s}$, cold white light in MC1000), which significantly inhibit growth of the non-mutated cells, indicating that the MMS-induced random mutagenesis could accelerate the adaptive evolution process as well as the hypermutation system (HS84). When the cells were transferred to harsher environments with further increased light intensities and temperature, the hypermutable cells (HS84) showed advantages and adapted to the conditions of 45°C and $2000 \mu\text{mol photons/m}^2/\text{s}$, while the MMS-treating group failed to survive. The advantages of the HS84 cells (with *in vivo* mutagenesis machinery) were supposed to result from continuous generation of genomic mutations in offspring populations, while in contrast, the genetic diversities in the MMS-treating group could not be further enriched without additional rounds of MMS treatment. We believed that the chemical or physical mutagenesis approaches could no doubt improve efficiency of ALE, while iterative rounds of manual manipulations would be necessary. In contrast, our engineered hypermutation system could be expected to simplify the process by facilitating continuous and autonomous *in vivo* mutagenesis. The above data and respective presentations have also been supplemented in the revised manuscript.

Fig. R3 Comparison of the engineered hypermutation system and MMS-induced random mutagenesis method on improving HLHT tolerance of *Synechococcus*. a. Diagram for the process of evolving HLHT tolerance with MMS-treated WT cells and hypermutable HS84 cells. The *Synechococcus* cells were inoculated in BG11 medium at an initial OD₇₃₀ of 0.2 in column photobioreactors and cultivated under 150 $\mu\text{mol photons/m}^2/\text{s}$ white fluorescent light at 30°C. WT cells (10^8 cells/mL) were incubated with 2% MMS for 1 minute. Cells were then washed with fresh BG11 twice. And then approximately 1×10^9 cells of HS84, WT, and WT treated with MMS were collected and plated onto solid BG11 medium and cultured under WT-lethal HLHT conditions (44°C and 500 $\mu\text{mol photons/m}^2/\text{s}$, warm white light). Meanwhile, HS84, WT, and WT treated with MMS were inoculated in BG11 medium at an initial OD₇₃₀ of 0.2 in column photobioreactors and cultivated in MC1000 under 42°C and 800 $\mu\text{mol photons/m}^2/\text{s}$. Then, cells cultured in the mid-log phase were inoculated in BG11 medium at an initial OD₇₃₀ of 0.05 in column photobioreactors and cultivated in MC1000 under 42°C and 1500 $\mu\text{mol photons/m}^2/\text{s}$.

After that, cells cultured in the mid-log phase were inoculated in BG11 medium at an initial OD₇₃₀ of 0.05 in column photobioreactors and cultivated in MC1000 under 42°C and 2000 μmol photons/m²/s and 45°C and 2000 μmol photons/m²/s. b. HLHT tolerance screening results of MMS-induced WT and HS84. c. Results of the evolution of HLHT tolerance.

3. Cultivation conditions are dubious. 500 mikro mol is not normal light intensities but already stressful. For this reason, authors probably used 150 mikromol for pre-cultures (line 624)?

RESPONSE: Thanks for the comments and we apologized for any confusion causing presentations about the cultivation conditions in the manuscript. When different light sources are used in microalgae cultivation, the promoting or inhibitory effects on cell growth can vary dramatically even when the same light intensities are set. In fact, the 500 μmol photons/m²/s light intensities were from cold white light sources equipped in MC1000, while the 150 μmol photons/m²/s light intensities were from white fluorescent light sources, and we have clarified the utilization of specific light sources in different sections of experiments in the manuscript (Cyanobacteria strains and culture conditions, Evaluation of relative mutation rates of *Synechococcus* genome replication, and Isolation of evolved *Synechococcus* strains with improved HLHT tolerance).

Regarding the cultivations in MC1000, we have performed evaluations about the *Synechococcus* cells' growth under different light intensities, and as shown in Fig. R4, cells cultivated under 500 μmol photons/m²/s light intensities were not inhibited but showed better growths than that under lower (300 μmol photons/m²/s light intensities) or higher (1000 μmol photons/m²/s light intensities) light densities. This result is also close to the phenomenon previously reported by another group (Ungerer, et al., reported that *Synechococcus* show optimal growth under light intensities of about 400 μmol photons/m²/s light intensities in MC1000 ⁷). Thus, it could be assumed that *Synechococcus* cells were not inhibited under 500 μmol photons/m²/s light intensities in MC1000.

Fig. R4 Growths of WT *Synechococcus* cells under 300, 500, 700, and 1000 μmol photons/m²/s at 30°C in MC1000.

Regarding the 150 $\mu\text{mol photons/m}^2/\text{s}$ light intensities mentioned in Line 624 of the previous version of the manuscript, it was used for cultivating the cells not in MC1000 but in our manually assembled cultivation system with white fluorescent light sources. This system, light source, and light intensities in this range (100 to 250 $\mu\text{mol photons/m}^2/\text{s}$) have been used for the cultivation and evaluation of *Synechococcus* strains and cell factories in our previous work⁸⁻¹⁰ and proven to effectively support cell growth and metabolism. Such a manually assembled system was adopted in the initial phase of this work for rapid and convenient cultivation and collection of *Synechococcus* cells. And when the temperature and light intensities for cell cultivation needed to be precisely set and controlled in the subsequent parts, the MC1000 system was used to maximize the accuracy of experimental conditions and the data reliabilities from the controls, parallels, and replicates.

4. Carefully check the text for ambiguous phrasing. There are many, and a part of them is listed below.

RESPONSE: Thanks for the suggestions we have checked and revised the whole manuscript.

Details:

2: Title. "acclimates". Conventionally, acclimation refers to physiological adjustments and adaptation to genetic adjustments. Title is misleading.

RESPONSE: Thanks for the comments. We have revised the title of this manuscript as "*Engineered hypermutation adapts cyanobacteria photosynthesis to combined high light and high temperature stress*", replacing "acclimates" with "adapts" to refer to the facts of genome mutagenesis and genetic adjustments in this work, and adding "combined" to refer the focus on HLHT tolerance as the target phenotype for improvement.

35. The authors mean simultaneous application of high light and high temperature. They should write it this way.

RESPONSE: Thanks for the suggestion. As mentioned in the previous replies, we have revised the title by adding the word "combined" to "high light and high temperature". In addition, we have supplemented more descriptions and presentations in the revised manuscript to emphasize the focus of this work on the photosynthesis tolerance to the simultaneous application effects of high light and high temperature stress (as seen in Line 60-81 of the revised manuscript).

36: Previous attempts might have been time-consuming but they were certainly not inefficient.

RESPONSE: Thanks for the comments. We fully accept the reviewer's critical comments regarding our not-appropriate and not-accurate presentations about the previous attempts and achievements in this area. We have modified the relative expressions throughout the manuscript as suggested. For example, in the abstract, it was revised as "laborious and time-consuming" (as seen in Line 36), in the introduction part, it was revised as "takes weeks or months" (as seen in Line 95), and in the discussion part, the "years of continuous cultivation" was revised as "long term continuous cultivation".

36: "phototrophs" wrong term here.

RESPONSE: Thanks for the comments and it has been corrected as "photoautotrophs" (as seen in Line 36).

38: cyanobacterial strain

RESPONSE: Thanks for the comments and it has been corrected as suggested (as seen in Line 38).

42: again "acclimation" when "adaptation" would be more meaningful

RESPONSE: Thanks for the suggestion and we have made the corrections as suggested (as seen in Line 42)

44: HIGH temperatures and light intensities

RESPONSE: Thanks for the comments and we have revised as suggested (as seen in Line 44)

45-47: Correct, but this was not the first study to show this.

RESPONSE: Thanks for the comments. We have rephrased the sentence as "The engineered hypermutation system expanded the cyanobacterial engineering toolbox and could be utilized for improving and deciphering the mechanisms leading to efficient and robust photosynthesis" (as seen in Line 46-47).

52: "most active"? in which sense?

RESPONSE: Thanks for the comments and we have removed "most active" from the sentence to avoid misleading presentations.

55: “artificial domestication”. The entire sentence is not really meaningful and the term “artificial domestication” makes little sense. Moreover, photosynthesis came for free during domestication.

RESPONSE: Thanks for the comments and we have removed the sentence from the revised manuscript to avoid mis-leading presentations.

58-60: needs rephrasing

RESPONSE: Thanks for the comments and we have rephrased the sentence as “Photosynthesis is performed by complex and sophisticated biochemical and molecular machineries, and high photosynthetic efficiency could only be achieved within an optimal range of physical and chemical parameters, while extreme and fluctuating environmental conditions would impair photosynthetic efficiency” (as seen in Line 60-63).

61: “high light and temperatures are major stresses that occur simultaneously”. This is not 100% right. Actually, plants are used to the simultaneous presence of LH and HT and shut down photosynthesis during these periods, also to avoid water loss. The problem is the combination of cold and high light, like happening in the morning of spring and autumn. This is really harmful since light reactions run hot and the dark reactions are still slow.

RESPONSE: Thanks for the comments. Firstly, we have rephrased this sentence as “High light and high temperatures are both severe environmental stresses that impair photosynthetic efficiency, and the two stresses would sometimes occur in coupling, causing superimposed effects” and we have also revised the relative presentations in the revised manuscript to explain the reason that we chose photosynthesis tolerance to combined high light and high temperature (HLHT) stress as the target phenotype (as seen in Line 66-81).

In addition, we totally agreed with the reviewer’s opinion that some other combined stresses such as “LT+HL” would also cause harmful and even more severe threats to the efficiency and stability of photosynthesis. And we hope that the improvement of cellular tolerance to such stresses and the understanding about the relative mechanisms could be promoted by utilizing our approach in combination with the existing methods in the future.

63: functional range of photosynthesis: what does this mean?

RESPONSE: Here the “functional range of photosynthesis” referred to the range of

light intensities and temperature that is optimal for photosynthesis activities. And we have rephrased the sentence as “Excessive exposure to solar energy and the resulting high temperature beyond the adaption range of photosynthesis machineries leads to the accumulation of intracellular reactive oxygen species (ROS)” (as seen in Line 69-70).

67/68: See above. I disagree. The study would be much more useful, if the two single components (HL or HT) and the double stress (HL + HT) had been analysed. Only analysing the combined stress is much less informative. Moreover, having studied HL separately would have allowed comparison with the previous study that the authors mention in *Synechocystis*.

RESPONSE: Thanks for the comments. As mentioned in previous replies, we agreed that the comparisons between evolved mutants and mechanisms tolerant to single stresses and the combined stress (e.g., HLHT versus HL or HT) would be very interesting and could provide much meaningful information about microbial & plant physiology, metabolism, and evolution. However, we believed that such experiments and relative analysis could form another interesting story and would be beyond the scope of this work. We have designed and initiated experiments for parallel evolution of tolerant phenotypes toward single stresses and combined stress with the newly developed engineered hypermutation system as well as other classical random mutagenesis approaches and adaptive laboratory evolution strategy. The experiments are currently underway and we hope that more systematic presentations could be given in the future.

In addition, some of the SNPs, functional genes, and mechanisms identified contributing to the HLHT tolerance were in consistence with the previously reported results about HL tolerant, and we have made relative discussions in the manuscript (as seen in Line 603-629).

70: “putative” can be deleted.

RESPONSE: Thanks for the suggestion. We have deleted the “putative” and revised the sentence as "Sharing a common ancestor with chloroplasts, cyanobacteria have similar photosynthetic mechanisms with eukaryotic microalgae and higher plants" (as seen in Line 80-83).

71-73. also algae have short life cycles and rapid growth.

RESPONSE: Thanks for the comments and we have rephrased the sentence as “Besides, cyanobacteria also have the characteristics of short life cycles, rapid growth

rates, and simple structures, allowing convenient genotype-phenotype mapping for photosynthetic research” (as seen in Line 85-87).

82: authors omitted to cite literature on HT ALE: Tillich et al. 2012 and 2014 for HT ALE.

RESPONSE: Thanks for the comments and we have added citations as suggested (as seen in Line 97).

83: ideal mutants. No, this is not ideal or if this is ideal then many more combinations are equally ideal.

RESPONSE: Thanks for the comments. We have removed the “ideal mutant” presentations and revised the text as “Moreover, mutants with improved tolerance to the combined stress of high light and high temperature are rarely reported; thus, the mechanisms of cyanobacterial photosynthesis tolerance to HLHT stress are still not as clearly understood as that to every single stress” (as seen in Line 97-100).

88: 10 to the three magnitude cell replication events. Rephrase. Unclear.

RESPONSE: Thanks for the comments. As suggested, we have rephrased the sentence as “it is estimated that it would take thousands of cell replication events to introduce one genetic mutation into the cyanobacteria genome ” to make more clear presentation (as seen in Line 105).

188: 4 days corresponds to how many generations? Is this adaptive evolution or just classical forward genetics in which single mutations are identified rather than epistatic series of mutations?

RESPONSE: Thanks for the comments. “4 days” refers to the time for obtaining tolerant colonies on the solid plates cultivated under lethal (to the WT control) HTHL conditions, thus it is difficult to estimate the number of generations elapsed in the process of growing from each single mutant cell (survived on the plate) to an observed colony (with different sizes). Besides, the *Synechococcus* cells (both WT and HS84) would be cultivated in non-lethal conditions for about 5 days before being spread on plates, and during this cultivation process, spontaneous mutations were expected to be generated on the genome of HS84 offspring cells (with a much higher rate than WT), expanding the genetic/phenotypic diversities and benefiting the obtaining of tolerant mutant in the subsequent step. Thus, the process shown in Fig. 3 of the manuscript could be recognized as classical forward genetics research. And we have also supplemented additional experiments showing that the engineered hypermutation

system could also be used in an ALE work to accelerate the process and save the requirements on iterative rounds of chemical/physical mutagenesis manipulations (as seen in Supplementary Fig. 9, as well as Fig. R3 in the previous response comments).

194: the authors selected three colonies and later many more (239-240). That might reflect the original history of the experiments but there is no need to write it this way because it is confusing at the end. This sound more like a lab book in this part of the manuscript.

RESPONSE: Thanks for the comments and we have revised the manuscript as suggested. Results and presentations about the genome re-sequencing and SNPs information have been summarized in the section of “Mining functional mutations from HLHT tolerant *Synechococcus strains*” (as seen in Line 274-337).

197: 30 C and 500 mikromol light cannot be considered as “normal conditions”.

RESPONSE: Thanks for the comments. As explained in the previous response (Fig. R4), 30°C and 500 $\mu\text{mol photons/m}^2/\text{s}$ (given by white cold light sources) in MC1000 would not inhibit the growth of *Synechococcus* cells. To avoid potential mis-leading effects, we have also used the NLNT (normal light intensities and normal temperature) to describe this condition.

Fig. R4 Growths of WT *Synechococcus* cells under 300, 500, 700, and 1000 $\mu\text{mol photons/m}^2/\text{s}$ at 30°C in MC1000.

243: heterozygous? Diploid organisms? What do you mean. Mention copy number of DNA.

RESPONSE: Thanks for the comments. *Synechococcus elongatus* PCC 7942 is polyploid and carries more than one copy (usually 5-10) of chromosome in each cell, and thus alleles carrying different SNPs could be harbored in the mutants. The

“heterozygous” status indicates that both the mutation and WT peaks could be identified from the sequencing reads in the re-sequencing experiments and the subsequent PCR-sequencing confirmations. We have also supplemented the explanations in the revised manuscript (as seen in Line 284-287).

244: what are repeated mutations?

RESPONSE: Thanks for the comments. The “repeated mutations” referred to the “mutations repeatedly occurred in more than one evolved strain”, to make more clear presentations, we have revised the manuscript (as seen in Line 287-288).

281: here and many times again. Acclimation is not adaptation and vice versa.

RESPONSE: Thanks for the comments. We have replaced “acclimation” with “adaptation” all throughout the revised manuscript as suggested.

335. scarifying. Rephrase.

RESPONSE: Sorry for the mis-spelling and we have rephrased the sentence as “indicating that the growth advantages imparted by the 1799 mutation under the HLHT conditions were obtained by sacrificing photosynthesis efficiencies under normal physiological conditions” (as seen in Line 380).

412: energy-coupled electron transfer between NADH and NADPH: explain.

RESPONSE: Thanks for the comments. H⁺ translocating NAD(P) transhydrogenase is a specific integral membrane protein complex and couples the simultaneous reduction of NADP⁺ and oxidation of NADH to the translocation of protons along the membrane proton gradient, which has a direct role in the regulation of electron carrier redox balance by catalyzing energy-coupled electron transfer between NAD(H) and NADP(H)^{11, 12}. We have revised the sentence as “conversion between NADH and NADPH might play an important role in regulating redox and energy balance of electron carriers in photosynthesis under HLHT conditions.” to make more clear presentations (as seen in Line 474).

472: this is not correct. Any ALE can be done in a couple of months, in particular if using fast-growing UTEX strains like the ones described recently by the Pakrasi lab.

RESPONSE: Thanks for the comments. We have revised the presentations by replacing “years of” with “long term” (as seen in Line 578). We agreed that the emerging fast-growing cyanobacteria strains could be used for a rapid and efficient ALE process. While it is also noteworthy that, as mentioned previously, no matter the

ALE process or classical direct screening process on solid plates, the generation of genome mutations would be a key pre-request to achieve genetic/phenotypic adjustment. Approaches to elevate replication mutation rates and accelerate the evolution process should also be adopted for evolving the fast-growing strain. The strategy and method developed in this work could also be expected to engineer and evolve such strains as well as the classical chemical/physical mutagenesis methods.

492: Previous study with HL tolerance in *Synechocystis* showed already that single mutations can do (part of) the job. So, this is not new.

RESPONSE: Thanks for the comments. As the reviewer pointed out, single mutations have proved important for remodeling cellular physiology and metabolism in cyanobacteria strains, and we have made the citations about the previous reports (as seen in Line 607). Here we were not claiming the “first report” or “first discovery” of such a phenomenon, but emphasizing that our findings further confirmed the “plasticities of the cyanobacteria genomes” in this work (as seen in Line 602).

536-541: is the overrepresentation of aromatic amino acids maybe due to the fact that they are hydrophobic and that photosynthetic proteins are mostly membrane proteins? Sometimes explanations are more obvious than thought. Make a control by analysing separately soluble and membrane proteins.

RESPONSE: Thanks for the constructive suggestions. We have made a separate analysis of the aromatic amino acids (Y, F, and W) proportions in total, soluble, and membrane proteins of *Synechococcus*. DeepTMHMM was used for the prediction of the topology of both alpha-helical and beta-barrel transmembrane proteins¹³. The 2657 proteins in *Synechococcus* were predicted and those with transmembrane domains were considered to be membrane proteins. 560 proteins were predicted to be transmembrane proteins and 2097 proteins were predicted to be soluble proteins (Supplementary Data 10). As shown in Supplementary Fig R5, the proportions of such amino acids are indeed higher than the average value in total proteins and that of the soluble proteins are relatively lower. In the photosystem, 34 proteins were predicted to be transmembrane proteins and 26 were predicted to be soluble proteins. However, the ratio of Y, F, and W in the photosystem was higher than the average value of transmembrane proteins, among which, all of the proteins in PSII were predicted to be transmembrane proteins, and the ratio of Y, F and W were much higher than the average value of transmembrane proteins. So even compared to transmembrane proteins, the proportion of these three aromatic amino acids in the photosystem is indeed relatively high, indicating that the hypothesis that enhanced shikimate pathway

gave guarantee for supplying abundant aromatic amino acids for photosynthesis machinery could not be simply excluded.

Fig. R5 Total frequencies of Y, F, and W of membrane proteins, soluble proteins, total proteins, and proteins in different components of photosynthesis and oxidative phosphorylation.

574-576: disagree. The approach might be faster (maybe not if hyper mutator has to be removed individually in each clone) but the approach is not new. At the end, the mutations do not differ from the ones found by ALE.

RESPONSE: Thanks for the comments. Firstly, as mentioned in the manuscript and previous response, the hypermutators could be restored through one round of genetic manipulation (transforming the mutant with a general vector) in about 7 days; and in fact, for mining functional SNPs and exploring the potential mechanisms, the mutator phenotype does not have to be restored.

Secondly, although the *in vivo* mutagenesis strategies and methods have been adopted for improving microbial phenotypes previously, we performed systematic screening for potential mutator/anti-mutator genes contributing to the regulation of replication fidelity in cyanobacteria for the first time. More importantly, we reported that environmental stress has significant synergistic effects with the genetic fidelity deficiency on triggering a hypermutation status, and that could be expected to benefit the optimization of existing systems in other organisms. Thus, we thought our work makes actual contributions to the methodology of microbial strain improvement.

The approach we developed in this work was not aiming to replace or compete with ALE or classical random mutagenesis methods, but to expand the toolbox for engineering and improving cyanobacteria, and it could be utilized to accelerate the ALE process or even in combination with the existing chemical/physical mutagenesis methods. For the HLHT tolerance improvement in this work, some previously reported SNPs (e.g., AtpA-C252Y) and some SNPs similar/related to previously reported ones

(e.g., NdhF) were indeed enriched and identified. But more importantly, some novel SNPs (primarily or assertively contributing to the improved HLHT tolerance; e.g., mutations located in NC2, 1831, 0189, and 0884) were also discovered in the evolved strains. Targeting the NC2 mutation, we reported for the first time that the enhanced shikimate pathway could remodel the photosynthesis chain, enhance the CEF activities and improve the efficiency and stability of *Synechococcus* photosynthesis, which could prove the effectiveness and significance of our approach.

Finally, we also appreciated the reviewer's comments and have revised the manuscript as "the elevated genome mutation rates and evolution rates of *Synechococcus* cells facilitated the identification of novel genetic evolutionary paths for cyanobacteria to adapt the HLHT stress".

987: normally you need DCMU to determine F_m . Could not find this in Methods.

RESPONSE: Thanks for the comments. The saturation pulse method associated with the pulse-amplitude-modulation technique, which is commonly used in the measurement of chlorophyll fluorescence measurements¹⁴, was applied in this research using a Dual-PAM 100 (Heinz Walz). At the reviewer's prompt, we realized that cyanobacteria might show different characteristics of chlorophyll fluorescence from those of land plants and manual calculation using the F_m level obtained by the addition of DCMU is necessary for the measurements of cyanobacterial cells¹⁵. Therefore, we recorded the maximum chlorophyll fluorescence after adding 20 μM DCMU with AL ($54 \mu\text{E m}^{-2}\text{s}^{-1}$) on as the maximum F_m . We have supplemented the descriptions in the revised manuscript (as seen in Line 859).

Reviewer #2 (Remarks to the Author):

NatComm

Title: Engineered hypermutation acclimates cyanobacteria photosynthesis to high light and high temperature stress

The manuscript of xx reports on improved mutation rates to capture cyanobacteria strains (*Synechococcus*) by identification and knockout of multiple genes involved in DNA replication fidelity. Moreover, the authors show that mutation rates can be further enhanced by exposing cultures to high light and/or high temperature, producing a large number of HLT strains. Many of the "HLT" mutants exhibited mutations causing enhanced expression of shikimate kinase, an enzyme of a major pathway for secondary metabolite production. Furthermore, the researchers overexpressed shikimate kinase in a *Synechocystis* strain, and showed that the overexpression strain

also gained tolerance to high light. They also suggest that upregulation of the shikimate pathway may represent a universal strategy for improving tolerance to light and/or temperature stress in cyanobacteria. Last, RNAseq and qPCR were performed to estimate the overall impact on global metabolism, photosynthesis and other regulatory processes. This work is a significant step forward in significantly improving the mutational rate in cyanobacteria and this tool should be of great interest to the scientific community in rapidly engineering novel strains with improved tolerance to a range of climate-relevant stressors.

However, I do think that the discussion relied heavily on the RNAseq results to speculate on a lot of physiological changes in the mutants. It would be nice to see some additional measurements to support the changes within the photosynthetic apparatus.

RESPONSE: Thanks for the complete and accurate description of this work and the positive comments. We appreciated the reviewer's suggestion and we have supplemented additional experiments for measuring the physiological changes of the HLHT tolerant mutant (HS199, carrying NC mutation) and the WT strain. The results revealed that the CEF, respiration, and whole chain oxygen evolution rates were all enhanced in HS199, which are consistent with the predictions made according to the transcriptome changes. Besides, we have also performed targeted proteome assays (PRM) to compare the changes in abundance of important proteins between HS199 and WT, which could also support understandings and verifications on the transcriptome changes.

Line 35 – I would suggest adding “combined” to emphasize that you are referring to the combined HL/HT stress (same comment throughout the manuscript). I also suggest that the acronym “HLT” is a little confusing since many publications use “LT” for low temperature – you could consider using “HLHT”

RESPONSE: Thanks for the suggestions. We appreciated the reviewer's suggestion and have added the word “combined” to the title, the abstract, and the main text of the revised manuscript. We have also revised the acronym of “combined high light and high temperature” as “HLHT” all throughout the revised manuscript.

Line 70 – why do you refer to cyanobacteria as “putative” ancestors of chloroplasts?

RESPONSE: Thanks for the comments and we have revised the presentations as “Sharing a common ancestor with chloroplasts, cyanobacteria have similar photosynthetic mechanisms with eukaryotic microalgae and higher plants” (as seen in Line 83-84).

Introduction – it would be nice to see a good rationale for why it specifically important to isolate high light/high temperature – tolerant mutants

RESPONSE: Thanks for the comments. We have supplemented more presentations about the reason we chose HLHT tolerance of photosynthesis as the target phenotype to improve and decipher. High light and high temperature are both severe environmental stresses that impair photosynthetic efficiency, and the two stresses would sometimes occur in coupling, causing superimposed effects. Excessive exposure to solar energy and the resulting high temperature beyond the adaption range of photosynthesis machineries leads to the accumulation of intracellular reactive oxygen species (ROS), which further induces photosystem impairments and disrupts cellular homeostasis, such as the maintenance and repair of protein activities. Combined high light and high temperature (HLHT) stress pose serious threats to agriculture and other photosynthesis derived economic activities and can happen more widely and frequently as the climate change¹⁶⁻²⁰; and thus, improving the tolerances of photoautotrophs to HLHT is of great significance. However, although the specific responsive or protective mechanisms of plants and algae toward single stress (high light or high temperature) have been extensively explored, understandings of combination of the two stresses (HLHT) are still relatively limited (as seen in Line 63-81 and Line 97-100 of the revised manuscript).

Line 84 – I don't think it's accurate to say that photosynthetic tolerance to HL and HT stress is poorly understood.

RESPONSE: We appreciated the reviewer's comments and have revised the sentence as "However, although the specific responsive or protective mechanisms of plants and algae toward single stress (high light or high temperature) have been extensively explored, understandings on tolerance mechanisms to the combination of the two stresses (HLHT) is still relatively limited." (as seen in Line 78-81 of the revised manuscript).

Line 399 – the RNA seq results suggesting remodeling of key photosynthetic electron transport chain components was very interesting, particularly those associated with energy homeostasis. Did the authors consider supporting these findings with some photosynthetic measurements, such as oxygen evolution, chlorophyll fluorescence, or P700 spectroscopy? I think some basic photosynthetic measurements (beyond chlorophyll estimations) would be a significant contribution to the impact of this manuscript.

RESPONSE: We appreciated the reviewer's suggestion and we have performed

measurements of photophysiology parameters with WT and HS199 (carrying NC2 mutation). The results revealed that the CEF, respiration, and whole chain oxygen evolution rates were all enhanced in HS199, which are consistent with the predictions made according to the transcriptome changes. Based on the newly obtained data, we have also revised the descriptions of the effects and mechanisms of NC2 mutation regulating physiology and metabolism in *Synechococcus* (as seen in “NC2 mutation remodeled the photosynthetic metabolism in *Synechococcus*” section in the revised manuscript).

Line 520 – The finding that a major enzyme of the shikimate pathway can confer significant increases in stress tolerance is very interesting. I would like to see more discussion – this paragraph is pretty superficial regarding the potential impacts on cellular physiology and stress tolerance.

RESPONSE: We appreciated the referee’s suggestion. We have revised the discussion section about the functional mechanism of enhanced shikimate pathway on efficiency and stability of photosynthesis in *Synechococcus* cells (as seen in Line 649 to Line 703 of the revised manuscript). Based on the data from transcriptome and physiology assays as well as the information from the previous publications, we supposed that the enhanced shikimate pathway could regulate the physiology and metabolism of *Synechococcus* cells through the following mechanisms: 1) to influence the abundances of down-stream metabolites (e.g., indole-3-acetate) which might have regulatory effects and cause changes on global transcriptions; 2) to remodel the photosynthesis chain by regulating the abundances of essential components (as revealed from transcriptome) and plastoquinone (a downstream metabolite of shikimate pathway); 3) to elevate CEF activities of the photosynthesis (a synergistic effect from changing plastoquinone pool, elevating abundances of NDH complex components, down-regulating PSII activities, and other potential mechanisms); 4) to guarantee the supply of aromatic amino acids (Y, W, and F) derived from shikimate pathway. Previous, there have been several reports about the changes in components or metabolites involved in the shikimate pathway in microalgae or plant mutants with changed CEF. Here in this work, we gave evidence that the enhanced CEF activities and the regulated photosynthesis efficiency/stability could be a result of the enhancement of the shikimate pathway.

Reviewer #3 (Remarks to the Author):

This manuscript describes the use of a creative approach to identify mutations in the

cyanobacterium *Synechococcus* sp. PCC 7942 that confer high light and temperature (HLT) stress tolerance. The authors use a strategy of perturbing the fidelity of DNA replication combined with environmental stress that increases ROS production, which allowed direct and rapid selection of mutants that survive lethal HLT stress in a process that is much faster than conventional adaptive laboratory evolution. They nicely showed that two types of mutations on their own could confer HLT tolerance, whereas others can improve HLT tolerance in combination. One of the two major mutations (AtpA-C252Y) had been found previously, but the other (NC2) affected expression of shikimate kinase. They convincingly showed that overexpression of shikimate kinase confers HLT tolerance, even in a different species of cyanobacterium, and that this has a large and pleiotropic effect on the transcriptome of *Synechococcus*.

Overall, I found the approach and results in this manuscript to be interesting and convincing.

Major comment:

1. The NC2 mutation was identified as a major route to HLT tolerance, however the molecular mechanism of how the upregulation of shikimate kinase via NC2 mutation confers HLT tolerance was not determined. The authors' speculation (lines 539-541) "that an enhanced supply of aromatic amino acids might promote the synthesis of essential photosystem components" is intriguing and supported by finding other mutations related to protein synthesis (i.e., mutations affecting GTP synthesis and EF-Tu) that further enhance the HLT tolerance of NC2 (Fig. 5). PSII repair via D1 protein turnover and replacement is known to be very important for photoprotection in cyanobacteria, so it was surprising that the authors did not test this experimentally. They could examine rates of PSII photoinhibition and recovery in the presence or absence of lincomycin, rates of D1 synthesis, and levels of D1 protein.

RESPONSE: Thanks for the accurate and comprehensive summary of this work and the positive and encouraging comments. We have recognized that the explanations about the NC2 mutation effects on regulating photosynthesis efficiency and stability of *Synechococcus* cells in the previous manuscript relied heavily on the RNAseq results, and thus we have performed additional physiology measurement with the WT and HS199 strain aiming to give more convincing hypothesis describing the functional mechanisms of NC2 mutation. The physiology measurement results revealed that the CEF, respiration, and whole chain oxygen evolution rates were all enhanced in HS199, which are in consistence with the predications made according to the transcriptome changes. And the enhanced CEF was supposed to play an essential role for guarantee cell survival and rapid growth under HLHT conditions. Regarding the PSII activities, we found surprisingly that the O₂ evolution through PSII was reduced by HS199, and

we supposed that the reduced PSII activities could also be partially a result from the enhanced CEF activities^{21,22}. Due to the limitations on research conditions (lack of the conditions for radioisotope labeling experiments) and COVID-19, we were not able to calculate the D1 synthesis rate in *Synechococcus* cells in the last months, but we performed targeted proteome assays (PRM) to compare the changes on abundances of important proteins (including D1 and other components in photosystem) between HS199 and WT, and it was revealed that abundance of D1 protein (PsbA1, *Synpcc7942_0424*) was significantly enhanced in HS199 which could be promoted by the enhanced CEF and ATP supply, and could support the repair of PSII system.

Minor comments:

2. Title: “adapts” would be preferable to “acclimates”, because this is a heritable change. This should also be changed in the main text.

RESPONSE: Thanks for the comments and we have revised the title as “*Engineered hypermutation adapts cyanobacteria photosynthesis to combined high light and high temperature stress*”. We have also revised the whole manuscript, replacing “acclimate” and “acclimation” with “adapt” and “adaptation” (as seen in Line 2).

line 52: Clarify what is meant by “most active”.

RESPONSE: Thanks for the comments and we have removed “most active” from the sentence to avoid mis-leading presentations (as seen in Line 53).

line 70: Cyanobacteria are not “a putative ancestor of chloroplasts”. Extant cyanobacteria share a common ancestor with chloroplasts.

RESPONSE: Thanks for the suggestion. We have deleted the “putative” and revised the sentence as “Sharing a common ancestor with chloroplasts, cyanobacteria have similar photosynthetic mechanisms with eukaryotic microalgae and higher plants” (as seen in Line 83-84).

The manuscript would benefit from copyediting to correct some instances of English usage. Here are just two examples, but there are others:

line 312: change “Complying” to “Consistent”

line 335: What is meant by “scarifying”?

RESPONSE: Thanks for the comments. We have made the revision as suggested (as seen in Line 356 and Line 380). And we have made copy checking the language revision all through the manuscript.

1. Stapley, J. et al. Adaptation genomics: the next generation. *Trends Ecol Evol* **25**, 705-712 (2010).
2. Stinchcombe, J.R. & Hoekstra, H.E. Combining population genomics and quantitative genetics: finding the genes underlying ecologically important traits. *Heredity (Edinb)* **100**, 158-170 (2008).
3. Schierenbeck, L. et al. Fast forward genetics to identify mutations causing a high light tolerant phenotype in *Chlamydomonas reinhardtii* by whole-genome-sequencing. *BMC Genomics* **16**, 57 (2015).
4. Yoshikawa, K. et al. Mutations in *hik26* and *slr1916* lead to high-light stress tolerance in *Synechocystis* sp. PCC6803. *Commun Biol* **4**, 343 (2021).
5. Dann, M. et al. Enhancing photosynthesis at high light levels by adaptive laboratory evolution. *Nat Plants* **7**, 681-695 (2021).
6. Tillich, U.M. et al. The Optimal Mutagen Dosage to Induce Point-Mutations in *Synechocystis* sp. PCC 6803 and Its Application to Promote Temperature Tolerance. *PLoS ONE* **7**, e49467 (2012).
7. Ungerer, J., Lin, P.C., Chen, H.Y. & Pakrasi, H.B. Adjustments to photosystem stoichiometry and electron transfer proteins are key to the remarkably fast growth of the cyanobacterium *Synechococcus elongatus* UTEX 2973. *mBio* **9** (2018).
8. Qiao, C. et al. Effects of reduced and enhanced glycogen pools on salt-induced sucrose production in a sucrose-secreting strain of *Synechococcus elongatus* PCC 7942. *Appl. Environ. Microbiol.* **84**, e02023-02017 (2018).
9. Qiao, Y., Wang, W. & Lu, X. Engineering cyanobacteria as cell factories for direct trehalose production from CO₂. *Metab Eng* **62**, 161-171 (2020).
10. Song, K., Tan, X., Liang, Y. & Lu, X. The potential of *Synechococcus elongatus* UTEX 2973 for sugar feedstock production. *Appl. Microbiol. Biotechnol.* (2016).
11. Kämäräinen, J. et al. Pyridine nucleotide transhydrogenase PntAB is essential for optimal growth and photosynthetic integrity under low-light mixotrophic conditions in *Synechocystis* sp. PCC 6803. **214**, 194-204 (2017).
12. Jackson, J.B. A review of the binding-change mechanism for proton-translocating transhydrogenase. *Biochimica et biophysica acta* **1817**, 1839-1846 (2012).
13. Hallgren, J. et al. DeepTMHMM predicts alpha and beta transmembrane proteins using deep neural networks. Preprint at <https://www.biorxiv.org/content/biorxiv/early/2022/04/10/2022.04.08.487609.full.pdf>.

14. Ungerer, J., Lin, P.-C., Chen, H.-Y. & Pakrasi, H.B. Adjustments to photosystem stoichiometry and electron transfer proteins are key to the remarkably fast growth of the cyanobacterium *Synechococcus elongatus* UTEX 2973. *mBio* **9**, e02327-02317 (2018).
15. Ogawa, T., Misumi, M. & Sonoike, K. Estimation of photosynthesis in cyanobacteria by pulse-amplitude modulation chlorophyll fluorescence: problems and solutions. *Photosynth Res* **133**, 63-73 (2017).
16. Allakhverdiev, S.I. et al. Heat stress: an overview of molecular responses in photosynthesis. *Photosynth Res* **98**, 541-550 (2008).
17. Roeber, V.M., Bajaj, I., Rohde, M., Schmulling, T. & Cortleven, A. Light acts as a stressor and influences abiotic and biotic stress responses in plants. *Plant Cell and Environment* **44**, 645-664 (2021).
18. Chen, Y.E. et al. Responses of photosystem II and antioxidative systems to high light and high temperature co-stress in wheat. *Environ Exp Bot* **135**, 45-55 (2017).
19. Szymanska, R., Slesak, I., Orzechowska, A. & Kruk, J. Physiological and biochemical responses to high light and temperature stress in plants. *Environ Exp Bot* **139**, 165-177 (2017).
20. Hussain, S. et al. Photosynthesis research under climate change. *Photosynth Res* **150**, 5-19 (2021).
21. Allen, J.F. Cyclic, pseudocyclic and noncyclic photophosphorylation: new links in the chain. *Trends Plant Sci* **8**, 15-19 (2003).
22. Alric, J. The plastoquinone pool, poised for cyclic electron flow? *Front Plant Sci* **6**, 540 (2015).

Reviewers' Comments:

Reviewer #1:

Remarks to the Author:

The authors have revised the manuscript according to the suggestions of the two reviewers. The suggestions of the two reviewers were extensive and to address them paid off for the authors. I still feel very unhappy that the authors changed two variables (compared to previous studies) in their adaptive evolution approach: type of mutagenesis and type of selection. I still think it would have been wiser to include also the same selection as done in the classical ALE for HL tolerance of *Synechocystis*. Actually, the authors changed also a third variable by using *Synechococcus* instead of *Synechocystis*.

However, I do not want to delay this procedure any longer. There are still minor issues of which I list some here. This is certainly not a complete list and the authors might consult colleagues for a final check of the manuscript for inconsistencies and ambiguous phrasing.

2: "cyanobacterial photosynthesis"

70: "Adapation range": in this context "Acclimation range" might be better

72: "...homeostasis by impairing maintenance and repair..." original phrasing sounds weird.

80: "...extensively explored" and what are the extensive references for this?

98: "rarely reported" and what are the rare references for this?

104-108: this passage is misleading and gives the impression that mutations and evolution take forever. Given the population sizes present already in a litre of culture, cyanobacterial evolution is just working fine. Therefore, rephrase.

284: "heterozygous". This is used for diploid organisms. The terminology for polyploids is different.

Please use the right term

458: "significantly higher" provide p-values

480: "... were also regulated": how?

481: "sequestration": you mean "fixation"?

501: Can one start a sentence with "And"?

501 "detection light" is this correct?

504: "significantly lower" see 458

518: "over-oxidation" less PSII, less reduction, less over-reduction stress?!

698: "overflowed electrons"? rephrase

795: "rifampin" rifampicin as used in other passages?

858: "measure intensity of 33 mikroE" many people consider this as absurdly high.

Reviewer #2:

Remarks to the Author:

The authors have thoroughly considered this reviewer's concerns. I am satisfied with the revised document.

Reviewer #3:

Remarks to the Author:

This revised manuscript has been improved by the inclusion of additional data and changes to the text, and the authors have satisfactorily addressed the comments from my previous review.

Point-to-point replies to the reviews of manuscript NCOMMS-22-28713A

REVIEWER COMMENTS

Reviewer #1 (Remarks to the Author)

General:

The authors have revised the manuscript according to the suggestions of the two reviewers. The suggestions of the two reviewers were extensive and to address them paid off for the authors.

RESPONSE: Thanks for the comments. We would like to express our gratitude for the thoughtful comments and constructive suggestions from all the reviewers, which are very helpful to improve the quality of the manuscript.

I still feel very unhappy that the authors changed two variables (compared to previous studies) in their adaptive evolution approach: type of mutagenesis and type of selection. I still think it would have been wiser to include also the same selection as done in the classical ALE for HL tolerance of *Synechocystis*. Actually, the authors changed also a third variable by using *Synechococcus* instead of *Synechocystis*.

RESPONSE: Thanks for the comments. We felt sorry for any unclear presentations of the methods evaluation and comparisons that might have caused confusions and misunderstandings. In this work, we developed and evaluated the engineered hypermutation system with *Synechococcus elongatus* PCC 7942, and thus we also treated this strain with chemical mutagen (MMS) for phenotypes improvement and compared the effects with that from the method developed in this work (in the revision). When we adopt the similar strategy (engineered hypermutation) in *Synechocystis* for improving physiological or metabolic traits (e.g., HL tolerance) in future work, we would undoubtedly compare the effects of the different methods/approaches in parallel experiments with *Synechocystis*. During the first-round revision, we have compared the effects of chemical mutagen treatment and engineered hypermutation on both elevating the spontaneous mutation rates and generating HLHT tolerant mutants. Both the two strategies showed advantages comparing with the WT/No-treatment control in the liquid medium passages process, and the engineered hypermutation system worked more effectively in the solid plate selection system (used in this work for obtaining HLHT tolerant mutants) as well as in the liquid cultivation (generally utilized in classical LAE) when transferred to more harsh conditions. We still want to express that the engineered hypermutation system was developed in this work aiming not to replace or compete with the current toolbox but to collaborate with them to simplify and

accelerate the cyanobacteria improvement process.

However, I do not want to delay this procedure any longer. There are still minor issues of which I list some here. This is certainly not a complete list and the authors might consult colleagues for a final check of the manuscript for inconsistencies and ambiguous phrasing.

RESPONSE: Thanks for the suggestions and we have checked and revised the whole manuscript.

2: "cyanobacterial photosynthesis"

RESPONSE: Thanks for the comments. We have made the revision as suggested (as seen in Line 2 of the revised manuscript).

70: "Adapation range": in this context "Acclimation range" might be better

RESPONSE: Thanks for the comments. We have made the revision as suggested (as seen in Line 66 of the revised manuscript).

72: "...homeostasis by impairing maintenance and repair..." original phrasing sounds weird.

RESPONSE: Thanks for the comments. We have made the revision as "...which further induces photosystem impairments, impairs the protein synthesis and repair activities, and disrupts intracellular homeostasis" (as seen in Line 68 of the revised manuscript).

80: "...extensively explored" and what are the extensive references for this?

RESPONSE: Thanks for the comments and we have removed "extensively" and added respective citations (as seen in Line 74 of the revised manuscript).

98: "rarely reported" and what are the rare references for this?

RESPONSE: Thanks for the comments and we have added relative citations (as seen in Line 91 of the revised manuscript).

104-108: this passage is misleading and gives the impression that mutations and evolution take forever. Given the population sizes present already in a litre of culture, cyanobacterial evolution is just working fine. Therefore, rephrase.

RESPONSE: Thanks for the suggestion. We have revised the sentence as "Based on

classical models and data, it can be estimated that it would take thousands of cell replication events to form one genetic mutation into the cyanobacteria genome. Although desired mutant could still be obtained from large-sized populations (from culture with increased cells density or volume), the process for selecting and identifying the mutations would be time and labor consuming.” (as seen in Line 100 of the revised manuscript).

284: “heterozygous”. This is used for diploid organisms. The terminology for polyploids is different. Please use the right term

RESPONSE: Thanks for the comments and suggestions. We have replaced the “heterozygous” with “heterogeneous/heterozygous” (as seen in Line 254, Line 256 and FIG 4 of the revised manuscript).

458: “significantly higher” provide p-values

RESPONSE: Thanks for the suggestion. We have provided *p*-value as suggested (as seen in Line 422 of the revised manuscript).

480: “... were also regulated”: how?

RESPONSE: Thanks for the comments. As discussed in the manuscript, transcriptions of several transcription factors were also detected in HS199 cells, indicating the rewiring of transcriptional regulation hierarchy, which might cause the changed transcription of genes in carbon fixation and metabolism network. Besides, the changed photosystem and energy metabolism would cause changed supply of energy and reducing power, ATP and NADPH, in the *Synechococcus* cells, which would also induce regulation of the carbon metabolism network to adapt the change.

481: “sequestration”: you mean “fixation”?

RESPONSE: Thanks for the comments and we have replaced “sequestration” with “fixation” (as seen in Line 446 of the revised manuscript).

501: Can one start a sentence with “And”?

RESPONSE: Thanks for the comments. We have made the revision and replaced “And” with “Additionally” (as seen in Line 465 of the manuscript).

501 “detection light” is this correct?

RESPONSE: Thanks for the comments. The strains were cultured under 42°C and 1000 $\mu\text{mol photons/m}^2/\text{s}$ or 30°C and 500 $\mu\text{mol photons/m}^2/\text{s}$ conditions and cells in

mid-exponential phase were sampled to measure the whole-chain O₂ evolution rate. Different light intensities were set during the measure process to obtain light response curves. We have rephrased the sentence to “Additionally, with gradually increased light intensities during detection (as seen in Line 466 of the manuscript), PSII activity of HS199 (peaking at 0.02 mg O₂ L⁻¹ OD₇₃₀⁻¹ s⁻¹) remained lower than that of WT (peaking at approximately 0.025 mg O₂ L⁻¹ OD₇₃₀⁻¹ s⁻¹)”.

504: “significantly lower” see 458

RESPONSE: Thanks for the suggestion. We have provided *p*-value as suggested (as seen in Line 469 of the revised manuscript).

518: “over-oxidation” less PSII, less reduction, less over-reduction stress?!

RESPONSE: Thanks for the suggestion and sorry for the mistake. We have made the correction (as seen in Line 483 of the revised manuscript).

698: “overflowed electrons”? rephrase

RESPONSE: Thanks for the suggestion. We have replaced “overflowed” with “excess” (as seen in Line 560 of the revised manuscript).

795: “rifampin” rifampicin as used in other passages?

RESPONSE: Thanks for the suggestion. We have replaced it with “rifampicin” (as seen in Line 724 of the revised manuscript).

858: “measure intensity of 33 mikroE” many people consider this as absurdly high.

RESPONSE: Thanks for the suggestion and sorry for the mistake. The measure light intensity we used was level 8 which is 19 μE m⁻²s⁻¹. We have made the correction (as seen in Line 791 of the revised manuscript).